# Genetic determinants of endophytism in the *Arabidopsis* root mycobiome

Fantin Mesny [1], Shingo Miyauchi [1,2], Thorsten Thiergart[1], Brigitte Pickel[1], Lea Atanasova [3,4], Magnus Karlsson [5], Bruno Hüttel [6], Kerrie W. Barry[7], Sajeet Haridas [7], Cindy Chen[7], Diane Bauer[7], William Andreopoulos[7], Jasmyn Pangilinan[7], Kurt LaButti [7], Robert Riley[7], Anna Lipzen[7], Alicia Clum[7], Elodie Drula[8,9], Bernard Henrissat[10], Annegret Kohler[2], Igor V. Grigoriev [7,11], Francis M. Martin [2,12✉] & Stéphane Hacquard [1,13✉]

The roots of *Arabidopsis thaliana* host diverse fungal communities that affect plant health and disease states. Here, we sequence the genomes of 41 fungal isolates representative of the *A. thaliana* root mycobiota for comparative analysis with other 79 plant-associated fungi. Our analyses indicate that root mycobiota members evolved from ancestors with diverse lifestyles and retain large repertoires of plant cell wall-degrading enzymes (PCWDEs) and effector-like small secreted proteins. We identify a set of 84 gene families associated with endophytism, including genes encoding PCWDEs acting on xylan (family GH10) and cellulose (family AA9). Transcripts encoding these enzymes are also part of a conserved transcriptional program activated by phylogenetically-distant mycobiota members upon host contact. Recolonization experiments with individual fungi indicate that strains with detrimental effects in mono-association with the host colonize roots more aggressively than those with beneficial activities, and dominate in natural root samples. Furthermore, we show that the pectin-degrading enzyme family PL1_7 links aggressiveness of endophytic colonization to plant health.

[1] Max Planck Institute for Plant Breeding Research, 50829 Cologne, Germany. [2] Université de Lorraine, Institut national de recherche pour l'agriculture, l'alimentation et l'environnement, UMR Interactions Arbres/Microorganismes, Centre INRAE Grand Est-Nancy, 54280 Champenoux, France. [3] Research division of Biochemical Technology, Institute of Chemical, Environmental and Biological Engineering, Vienna University of Technology, Vienna, Austria. [4] Institute of Food Technology, University of Natural Resources and Life Sciences, Vienna, Austria. [5] Forest Mycology and Plant Pathology, Swedish University of Agricultural Sciences, SE-75007 Uppsala, Sweden. [6] Max Planck Genome Centre Cologne, Max Planck Institute for Plant Breeding Research, Cologne, Germany. [7] U.S. Department of Energy Joint Genome Institute, Lawrence Berkeley National Laboratory, Berkeley, CA, USA. [8] INRAE, USC1408 Architecture et Fonction des Macromolécules Biologiques, 13009 Marseille, France. [9] Architecture et Fonction des Macromolécules Biologiques (AFMB), CNRS, Aix-Marseille Univ., 13009 Marseille, France. [10] Department of Biological Sciences, King Abdulaziz University, Jeddah, Saudi Arabia. [11] Department of Plant and Microbial Biology, University of California Berkeley, Berkeley, CA, USA. [12] Beijing Advanced Innovation Centre for Tree Breeding by Molecular Design (BAIC-TBMD), Institute of Microbiology, Beijing Forestry University, Tsinghua East Road Haidian District, Beijing, China. [13] Cluster of Excellence on Plant Sciences (CEPLAS), Max Planck Institute for Plant Breeding Research, 50829 Cologne, Germany. ✉email: francis.martin@inrae.fr; hacquard@mpipz.mpg.de

Roots of healthy plants are colonized by a rich and diverse community of microbes (i.e. bacteria and fungi) that can modulate plant physiology and development[1–5]. Root colonization by arbuscular mycorrhizal, ectomycorrhizal and ericoid mycorrhizal fungi play fundamental roles in shaping plant evolution, distribution, and fitness worldwide[6–11]. In contrast, the physiological relevance of root mycobiota members that do not establish symbiotic structures, but have the ability to colonize roots of asymptomatic plants in nature remains unclear. These fungal endophytes are predominantly Ascomycetes[12,13], which can either represent stochastic encounters or engage in stable associations with plant roots[14–18]. Multiple factors driving the assembly of endophytic fungal communities have been identified, including climatic conditions, soil properties, species identities of the host and surrounding plants and abiotic stresses[12–14,16,18–22]. Re-colonization experiments with individual fungal isolates and germ-free Brassicaceae plants—non-mycorrhizal and previously reported as hosting root endophytes colonizing a broad range of hosts[23]—revealed various effects of mycobiota members on plant performance, ranging along the parasitism-to-mutualism continuum[2,24–26]. Importantly, the outcome of the interaction on plant health can be modulated by host genetics, host nutritional status, and local environmental conditions[27–30].

While the ectomycorrhizal lifestyle was shown to have arisen independently multiple times from saprotrophic ancestors—by convergent transposon-mediated genomic expansions and simultaneous losses of plant cell wall-degrading enzymes (PCWDEs)[31,32], some phylogenetically distant evolutionary trajectories to root endophytism have been described, from pathogenic[28,33,34] or saprotrophic ancestors[35]. Although genomic signatures of endophytism remain to be identified, these studies pinpointed that no contraction of PCWDE arsenals occurred during transitions to endophytism[25]. Genomes of dark-septate endophytes were shown to be enriched in genes encoding PCWDEs—but also aquaporins, secreted peptidases, and lipases—, in comparison to closely related fungi with other lifestyles[36]. Importantly, PCWDE-encoding genes were reported to be over-expressed during root colonization by diverse fungal endophytes[27,28,37], suggesting they might be key determinants of endophytism. Genetic factors underlying the endophytic lifestyle could however be multiple, and also niche- and host-dependent.

Here, we aim at better characterizing the evolution and function of the root mycobiota, by studying a diverse set of 41 cultured fungi that colonize roots of the non-mycorrhizal plant A. thaliana. Using comparative genomics and transcriptomics in combination with plant recolonization experiments, we identified genomic determinants underlying the endophytic lifestyle. Our results suggest that repertoires of PCWDEs of the A. thaliana root mycobiota are key determinants of endophytism, shaping fungal endosphere assemblages and modulating host fitness.

## Results

### Cultured isolates are representative of wild A. thaliana root mycobiomes. Fungi isolated from roots of healthy A. thaliana represent either stochastic encounters or robust endosphere colonizers. From a previously established fungal culture collection obtained from surface-sterilized root fragments of A. thaliana and relative Brassicaceae species[2], we identified 41 isolates that could be distinguished based on their rDNA internal transcribed spacer 1 (ITS1) sequences, representing 3 phyla, 26 genera, and 38 species of the fungal root microbiota (Fig. 1a). We first tested whether these phylogenetically diverse isolates were representative of naturally occurring root-colonizing fungi. Direct comparison with rDNA ITS1 sequence tags from a continental-scale survey of the A. thaliana root mycobiota[18] revealed that most of the

matching sequences were abundant (mean relative abundance, mean RA > 0.1%, 30 out of 41 strains), prevalent (sample coverage >50%, 22 out of 41), and enriched (root vs. soil, log2FC, Mann–Whitney U test, FDR < 0.05, 26 out of 41) in A. thaliana root endosphere samples at a continental scale (Fig. 1a). Quantitatively similar results were obtained using sequence data from the independent rDNA ITS2 locus (Spearman; Sample coverage: rho = 0.65, P < 0.01; RA: rho = 0.59, P < 0.01; Fig. 1b). The cumulative RA of the sequence tags corresponding to these 41 fungi accounted for 35% of the total RA measured in root endosphere samples across European sites[18], despite the under-representation of abundant Agaricomycetes and Dothideomycetes taxa (Fig. 1c). We next assessed the worldwide distribution and prevalence of these fungal taxa across 3,582 root samples from diverse plants retrieved from the GlobalFungi database[38]. Continent-wide analysis revealed that the proportion of samples with positive hits was greater in Europe (sample coverage: up to 30%, median = 4%) than in North America (sample coverage: up to 10%, median = 0.5%), and largely insignificant in samples from other continents (Fig. 1a). Interestingly, only a few of these 41 isolates were detected in leaves of A. thaliana at two locations in Germany (data re-analyzed from ref. [39], n = 51 samples), as well as in 2 647 leaf samples retrieved from the GlobalFungi database[38] (Supplementary Fig. 1). Results indicate that most of the cultured A. thaliana root colonizing fungi reproducibly and predominantly colonize plant roots across geographically distant sites irrespective of differences in soil conditions and climates.

### Root mycobiota members evolved from ancestors with diverse lifestyles. Given the broad taxonomic diversity of A. thaliana root mycobiota members, endosphere colonization capabilities may have evolved multiple times independently across distinct fungal lineages. We sequenced the above-mentioned 41 fungal genomes using PacBio long-read sequencing and annotated them with the support of transcriptome data (Methods), resulting in high-quality genome drafts (number of contigs: 9–919, median = 63; L50: 0.2–9.1 Mbp, median = 2.3 Mbp; Supplementary Data 1). Genome size varied between 33.3 and 121 Mb (median = 45 Mbp) and was significantly correlated with the number of predicted genes (number of genes: 10,414–25,647, median = 14,777, Spearman rho = 0.92, P = 3.82e−17) and the number of transposable elements (Spearman rho = 0.86, P = 4.13e−13) (Supplementary Fig. 2). A comparative genome analysis was conducted with 79 additional representative plant-associated fungi with previously well-described lifestyles[40], selected in the same or closely related phylogenetic classes as the strains we sequenced. Since classifying species into unique lifestyle categories is restrictive and can introduce bias[41], both the isolation of strains and previous knowledge about their species were considered to select plant pathogens, soil/wood saprotrophs, ectomycorrhizal symbionts, ericoid mycorrhizal symbionts, orchid mycorrhizal symbionts and endophytes[28,30,34,36,42–45] (Fig. 2a, Supplementary Fig. 3 and 4, Supplementary Data 2). Arbuscular mycorrhizal fungi were excluded from the study, as they are phylogenetically distant to the strains we isolated. To decipher potential evolutionary trajectories within this large fungal set, we first defined copy numbers of gene families in the 120 fungal genomes based on orthology prediction (n = 41,612; OrthoFinder[46]) and subsequently predicted the ancestral genome content using the Wagner parsimony method (Count[47]). Next, we trained a Random Forest classification model linking gene family copy numbers to lifestyles, resulting in a lifestyle prediction accuracy of $R^2 = 0.70$ (Methods). Although this classifier cannot confidently assign a single lifestyle to one genome content, it can be used to estimate lifestyle probabilities, and can reveal potential evolutionary

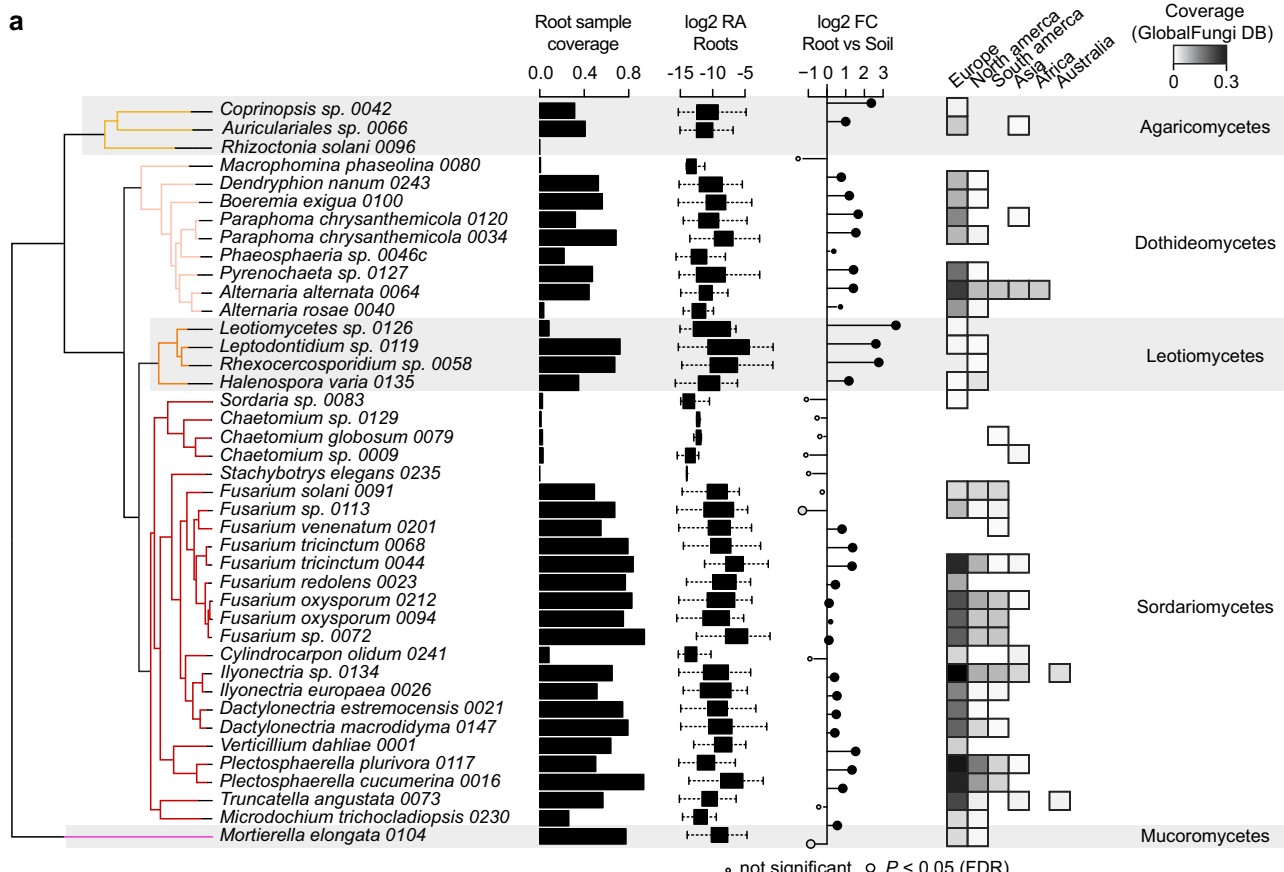

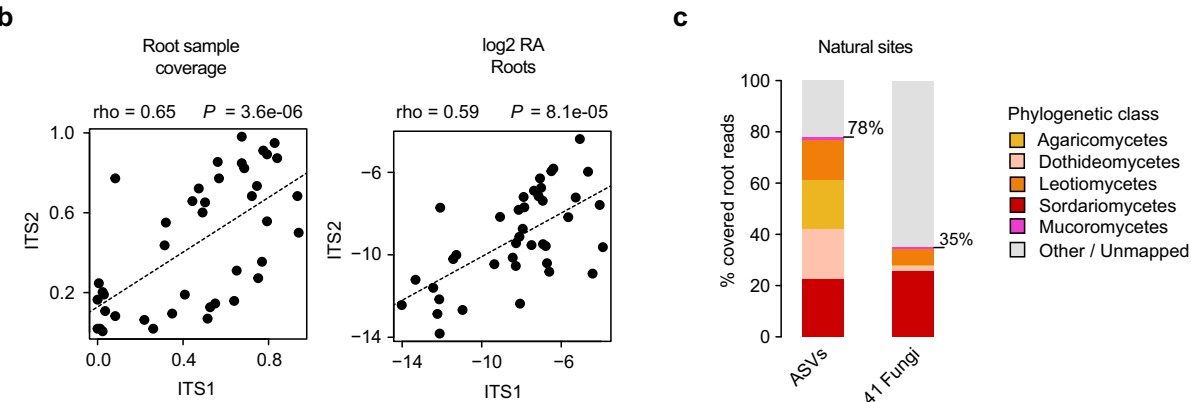

**Fig. 1 Prevalence and abundance profiles of 41 root-colonizing fungi across naturally occurring *A. thaliana* root mycobiomes. a** Species names and phylogenetic relationships among the 41 selected fungi. Estimated prevalence (i.e., root sample coverage, bar-plots), relative abundance (RA, log2 transformed, box-plots), and enrichment signatures (log2FC, circles) were calculated for each fungus based on data from a previously published continental-scale survey of the *A. thaliana* root mycobiota[18]. ITS1 tags from natural site samples were directly mapped against the reference ITS1 sequences of the selected fungi. Sample coverage in roots was computed based on $n = 169$ root samples and estimated RA were calculated for root samples having a positive hit only. On the RA boxplot, boxes are delimited by first and third quartiles and whiskers extend to show the rest of the distribution. Log2Fold-Change (log2FC) in RA between root ($n = 169$) and soil samples ($n = 223$) is shown based on the mean RA measured across samples and significant differences are indicated by circle sizes (two-sided Mann–Whitney $U$ test, FDR < 0.05, see detailed values in Supplementary Data 1). ITS1 sequence coverage measured across 3 582 root samples retrieved from the GlobalFungi database[38]. Note that samples were analyzed separately by continent. **b** Correlation between root sample coverage (left panel) measured in ITS1 ($n = 169$) and ITS2 ($n = 158$) datasets for each of the 41 fungi ($n = 41$, Spearman's rank correlation). Right panel: same correlation but based on log2 RA values ($n = 41$, Spearman's rank correlation). **c** RA profiles of naturally occurring fungi (class level) detected in *A. thaliana* roots across 17 European sites[18] ("all ASVs", left) and the corresponding distribution of the ITS1 sequences of the 41 selected fungi ("41 fungi", right). Note that the cumulative RA of the 41 fungi accounts for 35% of all sequencing reads detected in *A. thaliana* roots across European sites.

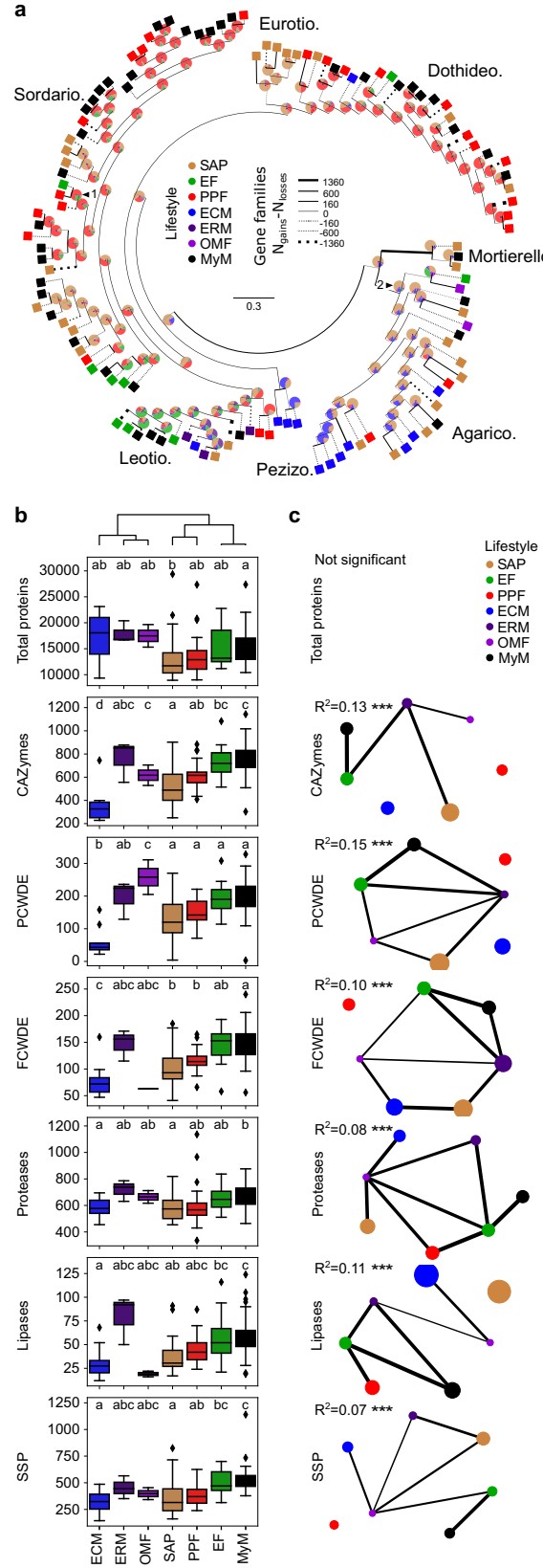

**Fig. 2 Ancestral relationships and trait convergence across root-colonizing fungal endophytes. a** Lifestyle-annotated whole-genome phylogeny of the 41 selected mycobiota members (MyM, black) and 79 published fungal genomes (SAP saprotrophs, EF endophytic fungi, PPF plant pathogenic fungi, ECM ectomycorrhiza, ERM ericoid mycorrhiza, OMF orchid mycorrhizal fungi). Pie charts on ancestor nodes show lifestyle probabilities of each ancestor, as identified by a Random Forest model trained on 79 non-mycobiota genome compositions in gene families ($R^2 = 0.70$). Two arrows highlight ancestral lifestyle predictions which corroborate previous reports: (1) the pathogenic ancestor of the endophyte *Colletotrichum tofieldiae* (2) the saprotrophic ancestor of ectomycorrhizal fungi and Sebacinales. Branch width is proportional to the gene family gains-losses difference ($N_{gains} − N_{losses}$). Line is dotted when this difference is negative. **b** Genomic counts ($n = 120$) of genes involved in fungal-host/environment associations (CAZymes carbohydrate-active enzymes, PCWDEs plant cell-wall degrading enzyme, FCWDEs fungal cell-wall degrading enzyme, SSPs small secreted proteins; PCWDEs and FCWDEs are CAZyme subsets). Boxes are grouped according to UPGMA hierarchical clustering on mean counts over the different categories. They are delimited by first and third quartiles, central bars show median values, whiskers extend to show the rest of the distribution, but without covering outlier data points (further than 1.5 interquartile range from the quartiles, and marked by lozenges). ANOVA-statistical testing (Counts~PhylogenyPCs+Lifestyle, Methods) identified both phylogeny and lifestyles as having an effect on genomic contents. The letters highlight the result of a two-sided post hoc TukeyHSD test that compares count differences exclusively due to the lifestyle. **c** Networks showing the results of a PERMANOVA-based comparison of gene repertoires (JaccardDistances~Phylogeny+Lifestyle, see Supplementary Data 3 for detailed $R^2$ and *P*-values). Networks for each category are labeled with Lifestyle $R^2$ values. ***$P < 0.001$ (Supplementary Fig. 6). Lifestyles are connected if their gene compositions are not significantly different. Node size is proportional to the area of one lifestyle's ordination ellipse on a Jaccard-derived dbRDA plot constrained by lifestyles, and reflects the intra-lifestyle variability. Edge weights and widths are inversely proportional to the distance between ordination ellipse centroids.

Agaricomycetes—were predicted to be saprotrophs[25,48] (see arrows numbered 1 and 2 on Fig. 2a). According to the classifier's predictions, Agaricomycetes and Mortierellomycetes in *A. thaliana* mycobiota likely derived from soil saprotrophs, while those belonging to Dothideomycetes and Sordariomycetes were predicted to have evolved from pathogenic ancestors. The ancestral lifestyle of Leotiomycete mycobiota members remains uncertain and could be multiple (Fig. 2a). Although the composition of our data set might influence these ancestral lifestyle predictions, our results nonetheless suggest that in planta accommodation of *A. thaliana* root mycobiota members occurred multiple times independently during evolution, as these fungi evolved from ancestors with diverse lifestyles.

**Functional overlap in genomes of root mycobiota members and endophytes.** Isolation of mycobiota members from roots of healthy plants prompted us to test whether their gene repertoires more extensively resemble those of mycorrhizal symbionts, known endophytes, saprotrophs, or pathogens. While the genomes of ectomycorrhizal fungi were shown to be enriched in transposable elements[31,32], the percentage of these elements remained low in genomes of root mycobiota members (0.69–28.43%, median = 5.44%, Supplementary Fig. 5). We annotated genes known to play a role in fungus-host interactions (Methods), including those encoding carbohydrate-active enzymes (CAZymes), proteases, lipases, and effector-like small secreted proteins (SSPs[49]), and then assessed differences in repertoire diversity across lifestyles (Fig. 2b). Unlike

trajectories when applied to Wagner-predicted ancestral genomic compositions (see pie charts, Fig. 2a). This probabilistic approach corroborated that recent ancestors of the beneficial root endophyte *Colletotrichum tofieldiae* were likely pathogenic[28], whereas those of beneficial Sebacinales—like those of ectomycorrhizal

ectomycorrhizal fungi[31,32], but similarly to endophytes[27,28,30,34,36], the genomes of root mycobiota members retained large repertoires of genes encoding PCWDEs, SSPs, and proteases (ANOVA-TukeyHSD, $P < 0.05$, Fig. 2b). Using permutational multivariate analysis of variance (PERMANOVA) and distance-based redundancy analyses (dbRDA)—based on Jaccard dissimilarity indices between genomes calculated on the copy numbers of genes in each family—, we distinguished lifestyle from phylogenetic signals in gene repertoire composition (Fig. 2c, Supplementary Fig. 6a). This revealed that "lifestyle" significantly contributes to the variation in gene repertoire composition (phylogeny: $R^2$: 0.17–0.46, $P < 0.05$; lifestyle: $R^2$: 0.07–0.15, $P < 0.05$, Supplementary Data 3). Interestingly, the factor "lifestyle" explained the highest percentage of variance for PCWDE repertoires (phylogeny: $R^2 = 0.26$; lifestyle: $R^2 = 0.15$, Supplementary Data 3), suggesting that these CAZymes play an important role in lifestyle differentiation. Further pairwise comparisons between lifestyle groups revealed that gene repertoire composition of root mycobiota members could not be differentiated from those of endophytes (post hoc pairwise PERMANOVA, $P > 0.05$, Fig. 2c). Therefore, gene repertoires of *A. thaliana* root-colonizing fungi resemble those of endophytes more than saprotrophs, pathogens or mycorrhizal symbionts. Across the tested gene groups, the families which contribute the most in segregating genomes by lifestyles (Supplementary Fig. 6b, Methods) include two xylan esterases (CE1, CE5), two pectate lyases (PL3_2, PL1_4), one pectin methyltransferase (CE8), and one serine protease (S08A). Further analysis focusing on total predicted secretomes (Supplementary Fig. 7, Supplementary Fig. 8a) and CAZyme subfamilies (Supplementary Fig. 8b) confirmed strong genomic similarities between *A. thaliana* root mycobiota members and known endophytic fungi.

**Genomic traits of the endophytic lifestyle**. To identify unique genetic determinants characterizing both known endophytes and *A. thaliana* root mycobiota members, the 120 genomes were mined for gene families whose copy numbers allow efficient segregation of these fungi ($n = 50$) from those with other lifestyles ($n = 70$). We trained a Support Vector Machines classifier with Recursive Feature Elimination (SVM-RFE) on the gene counts of orthogroups significantly enriched or depleted between these two groups (ANOVA, FDR < 0.05). A minimal set of 84 gene families that best segregated the two lifestyle groups was retained in the final SVM-RFE classifier ($R^2 = 0.80$, Fig. 3a and Supplementary Data 4a). These orthogroups can explain lifestyle differentiation independently from phylogenetic signal (PhyloGLM[50] – 83/84, FDR < 0.05) and were significantly enriched in enzymes (i.e., GO *catalytic activity*, GOATOOLS[51] FDR = 0.002, Supplementary Data 4b) and in CAZymes (one-sided Fisher Exact Test, odds ratio = 7.45, $P = 0.03$). Notably, genes encoding PCWDEs acting on pectin (CE12, GH145, PL11), cellulose (AA9), and hemicellulose (i.e., xylan: GH10, GH16, CE1) were identified, together with others encoding peptidases, transporters and proteins involved in amino acid metabolism (Fig. 3b and Supplementary Data 4a). These 84 gene families were analyzed for co-expression in published fungal transcriptomic datasets gathered in the database STRING[52]. An MCL-clustered co-expression network built on families enriched in known endophytes and *A. thaliana* mycobiota members revealed six clusters of co-expressed genes (Fig. 3c), including carbohydrate membrane transporters, and genes involved in carbohydrate metabolism (e.g., GH10) and amino acid metabolism. These functions are likely to be essential for endophytic root colonization.

**Root colonization capabilities explain fungal outcome on plant growth**. Root-colonizing fungi can span along the endophytism-parasitism continuum[25,53]. Consistently, our previously trained Random Forest lifestyle classifier ($R^2 = 0.70$, Fig. 2a) predicted

our 41 mycobiota members to be either plant pathogens, endophytes or saprotrophs (Fig. 4a). We tested the extent to which the 41 fungi can modulate host physiology by performing binary interaction experiments with germ-free *A. thaliana* plants grown in two nutrient conditions under laboratory conditions (inorganic orthophosphate, Pi: 100 μM and 625 μM $KH_2PO_4$, Fig. 4a). We identified that seed inoculation with the independent isolates influenced both germination rate (GR, Supplementary Fig. 9) and shoot fresh weight (SFW) of four-week-old plants (n = 7127), and therefore calculated a plant performance index (PPI = SFW * GR, Methods). Under Pi-sufficient conditions, 39% of the isolates (16/41) negatively affected host performance compared to germ-free control plants, whereas 61% (25/41) had no significant effect on PPI (Kruskal–Wallis–Dunn Test, *adj.* $P < 0.05$, Fig. 4a). Fungal-induced change in PPI was significantly modulated by the nutritional status of the host, as depletion of bioavailable Pi in the medium was associated with a reduction in the number of fungi with pathogenic activities (20%, 8/41) and an increase of those with beneficial activities (12%, 5/41) (Kruskal–Wallis–Dunn Test, *adj.* $P < 0.05$, Fig. 4a). Notably, PPI measured for low and high Pi conditions was negatively correlated with strain RA in roots of European *A. thaliana* populations (Spearman, High Pi: rho = −0.33, $P = 0.033$; Low Pi: rho = −0.49, $P = 0.0014$, Fig. 4b), suggesting a potential link between the ability of a fungus to efficiently colonize roots and the observed negative effect on plant performance. Consistent with this hypothesis, fungal load measured by quantitative PCR in roots of four-week-old *A. thaliana* colonized by individual fungal isolates (Supplementary Fig. 10ab), was positively correlated with fungal RA in roots of natural populations (Spearman, High Pi: rho = 0.57 $P = 0.0002$; Low Pi: rho = 0.52, $P = 0.0008$, Fig. 4c), and was also negatively linked with PPI outcome (Spearman, High Pi: rho = −0.44, $P = 0.005$, Low Pi: rho = −0.30, $P = 0.057$) (Supplementary Fig. 10cd). Furthermore, a co-occurrence matrix based on the RA of ASVs corresponding to these isolates in naturally occurring root mycobiomes indicated that most taxa with neutral and detrimental effects often co-occurred in roots of European *A. thaliana* populations[18], whereas those with beneficial activities were rarely detected (Supplementary Fig. 11). Taken together, our results suggest that robust root colonizers have a high pathogenic potential, and that their colonization must be tightly controlled not to affect plant health.

**A conserved set of CAZyme-encoding genes is induced in planta by diverse root mycobiota members**. We tested whether putative genomic determinants of endophytism defined above by a machine learning approach were part of a core response activated in planta by root mycobiota members. Six representative fungi from three different phylogenetic classes were selected for in planta transcriptomics on low Pi sugar-free medium: *Chaetomium sp*. 0009 (*Cs*), *Macrophomina phaseolina* 0080 (*Mp*), *Paraphoma chrysantemicola* 0034 (*Pc*), *Phaeosphaeria sp*. 0046c (*Ps*), *Truncatella angustata* 0073 (*Ta*), *Halenospora varia* 0135 (*Hv*). Confocal microscopy of roots grown in mono-association with these fungi highlighted similar colonization of root surfaces and local penetrations of hyphae in epidermal cells (Supplementary Fig. 12). After mapping of RNA-seq reads on genome assemblies (Hisat2[54]) and differential expression analysis (in planta vs. on medium, DESeq2[55]), significant log2 fold-change (log2FC) values were summed by orthogroups, allowing between-strain transcriptome comparisons (Methods). Transcriptome similarity did not fully reflect phylogenetic relationships since *Cs* and *Ta* (Sordariomycetes) clustered with *Hv* (Leotiomycete), whereas *Mp*, *Pc* and *Ps* (Dothideomycetes) showed substantial transcriptome differentiation (Fig. 5a). Although in planta transcriptional reprogramming was

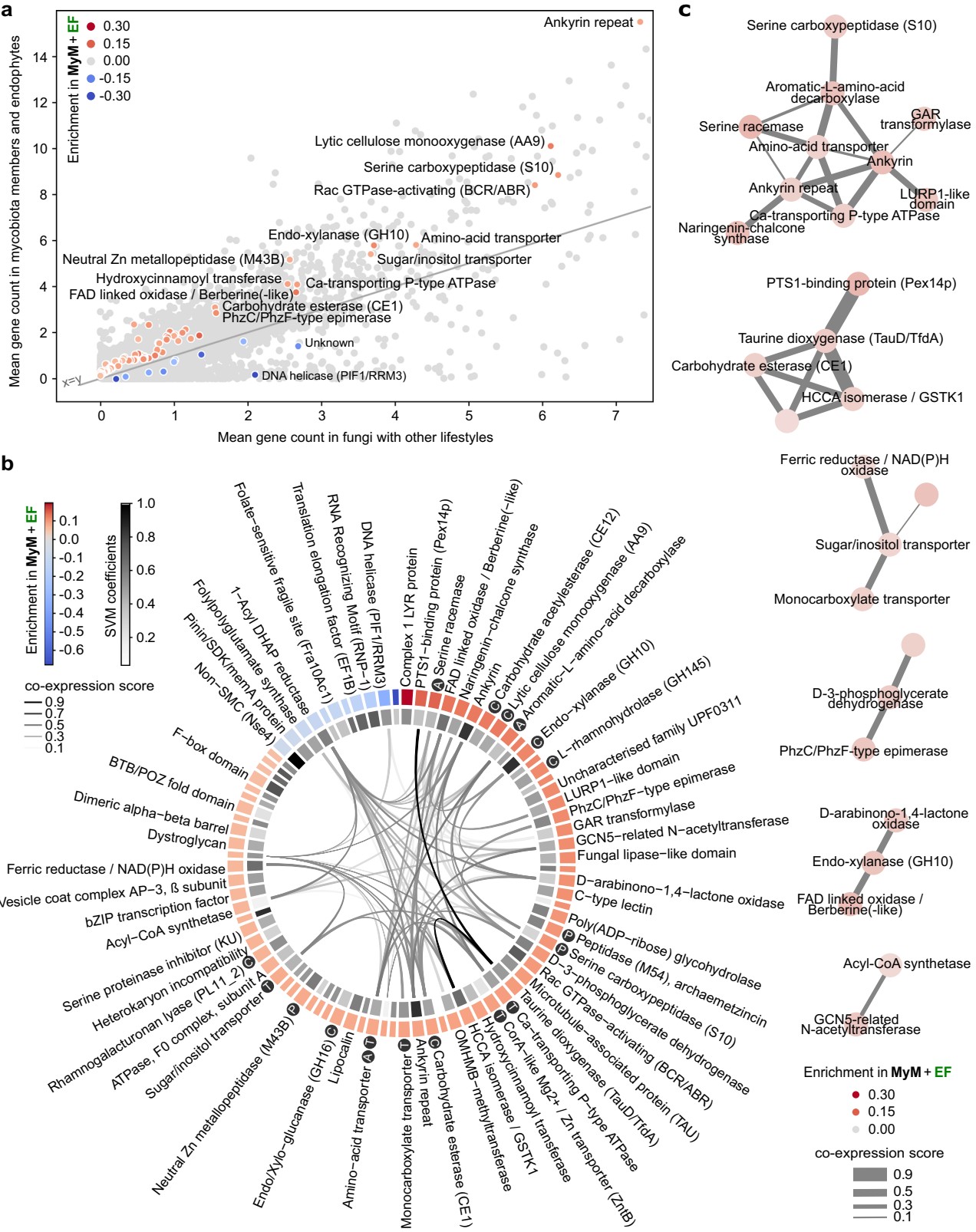

largely strain-specific, we identified a core set of 26 gene families that were consistently over-expressed by these distantly related fungi in *A. thaliana* roots (Fig. 5b). We observed a remarkable over-representation of genes coding for CAZymes acting on different plant cell wall components (i.e., 19/26, 73%), including cellulose, xylan and pectin (Fig. 5c). This set was also significantly enriched in

families previously identified as putative determinants of endo-phytism by our SVM-RFE classifier (Fisher exact test, $P < 0.05$), including AA9 (lytic cellulose monooxygenase) and GH10 (xyla-nase) CAZyme families. Inspection of fungal genes over-expressed in planta by each strain (Supplementary Data 5), followed by independent GO enrichment analyses, corroborated that

**Fig. 3 Minimal set of 84 gene families discriminating mycobiota members and endophytes from other lifestyles. a** Scatterplot showing the mean per-genome copy number of each orthogroup in mycobiota members and endophytes, in comparison to other lifestyles. Light gray: all 41,612 orthogroups. The 84 discriminant orthogroups identified by SVM-RFE ($R^2 = 0.8$) are highlighted in a gradient of red or blue colors reflecting, respectively, enrichment or depletion in *A. thaliana* mycobiota members and endophytes (MyM + EF) compared to the other fungal lifestyles. **b** Functional descriptions of the 84 discriminant orthogroups. This gene set is enriched in CAZymes (Fisher, $P < 0.05$, labeled C) and also contains peptidases (labeled P), transporters (labeled T) and proteins involved in amino-acid metabolism (labeled A). The outer circle shows orthogroup enrichment/depletion as described in panel a (see Supplementary Data 4a for associated ANOVA $P$-values). The inner circle depicts the SVM coefficients, reflecting the contribution of each orthogroup to lifestyle differentiation. In the center, links between orthogroups indicate coexpression of associated COG families in fungi (STRING database[52]). **c** Coexpression network of gene families across published fungal transcriptomic datasets, built on discriminant orthogroups enriched in endophytes and mycobiota members and clustered with the MCL method.

carbohydrate metabolic processes and xylanase activities were the most common fungal responses activated in planta (GOATOOLS, FDR < 0.05, Fig. 5d). Notably, we also observed important percentages of genes encoding effector-like SSPs induced in planta (9.8–42.4%, median = 21.6%). Together, these enzymes and SSPs are likely to constitute an essential toolbox for *A. thaliana* root colonization and for fungal acquisition of carbon compounds from plant material. Analysis of corresponding *A. thaliana* root transcriptomes revealed that different responses were activated by the host as a result of its interaction with these six phylogenetically distant mycobiota members (Supplementary Fig. 13, Supplementary Data 6). Our data suggest that phylogenetically distant mycobiota members colonize *A. thaliana* roots using a conserved set of PCWDEs and have markedly different impacts on their host.

**Polysaccharide lyase family PL1_7 as a key component linking colonization aggressiveness to plant health.** We reported above a potential link between aggressiveness in root colonization and detrimental effect of fungi on PPI. To identify underlying genomic signatures explaining this link, we employed three different methods. First, inspection of diverse gene categories across genomes of beneficial, neutral, and detrimental fungi revealed significant enrichments in CAZymes (especially polysaccharide/ pectate lyases, PLs) and proteases in the genomes of detrimental fungi (Low Pi conditions, Kruskal–Wallis $P < 0.05$, and Dunn tests, Supplementary Fig. 14a, b). In these categories, three pectate lyases (PL1_4, PL1_7, PL3_2) and three peptidases (S08A, A01A, S10) contributed the most in segregating genomes by effect on plants (see the count in gene copy in Supplementary Fig. 14c). Second, multiple testing of association between secreted CAZyme counts ($n = 199$ families in total) and fungal effect on PPI identified the PL1_7 family as the only family significantly linked to detrimental effects (ANOVA, Bonferroni; Low Pi: $P = 0.026$; High Pi: not significant; Fig. 6a). Finally, an SVM-RFE classifier was trained on the gene counts of all orthogroups that were significantly enriched or depleted in genomes of detrimental vs. non-detrimental fungi (ANOVA, FDR < 0.05). While this method failed at building a classifier to predict detrimental effects at high Pi (no families significantly enriched/depleted), it successfully predicted detrimental effects at low Pi with very high accuracy ($R^2 = 0.88$). A minimal set of 11 orthogroups discriminating detrimental from non-detrimental fungi was identified (Fig. 6b, Supplementary Data 7), and includes gene families encoding membrane transporters, zinc-finger domain-containing proteins, a salicylate monooxygenase and a PL1 orthogroup containing the aforementioned PL1_7 CAZyme subfamily and related PL1_9 and PL1_10 subfamilies. Further phylogenetic instability analysis based on duplication and mutation rates (MIPhy[56]) identified PL1_9 and PL1_10 as slow-evolving clades in the gene family tree (instability = 30.94 and 18.86 respectively, Fig. 6c), contrasting with most PL1_7 genes that were located in two rapidly evolving clades (index = 85.30 and 66.12). Of note, genomic counts of PL1_7, but not PL1_9/10, remained significantly associated to

detrimental host phenotypes after correction for the phylogenetic signal in our dataset (PhyloGLM[50], FDR = 0.03). PL1_7 was also part of the core transcriptional response activated in planta by six non-detrimental fungi (Fig. 5c) and was enriched in mycobiota members and endophytes in comparison to saprotrophs and mycorrhizal fungi (Supplementary Fig. 14d). Therefore, degradation of pectin by root mycobiota members is likely crucial for penetration of—and accommodation in—pectin-rich *A. thaliana* cell walls. However, the remarkable expansion of this gene family in detrimental compared to non-detrimental fungi predicts a possible negative link between colonization aggressiveness and plant performance. To test this hypothesis, we took advantage of the *Trichoderma reesei* QM9414 strain (WT, PL1_7 free background) and its corresponding heterologous mutant lines overexpressing *pel12*, a gene from *Clonostachys rosea* encoding a PL1_7 pectate lyase with direct enzymatic involvement in utilization of pectin[57]. By performing plant recolonization experiments at low Pi with these lines, we observed that *T. reesei pel12*OE lines negatively affected PPI with respect to their parental strain (ANOVA and TukeyHSD test, $P < 0.05$ for two out of three independent overexpressing lines, Fig. 6d), and this phenotype was associated with a significant increase in fungal load in plant roots (Kruskal–Wallis and Dunn test, $P < 0.05$, Fig. 6e). Taken together, our data indicate that pectin-degrading enzymes belonging to the PL1_7 family are key fungal determinants linking colonization aggressiveness to plant health.

## Discussion

We report here that genomes of fungi isolated from roots of healthy *A. thaliana* harbor a remarkable diversity of genes encoding secreted proteins and CAZymes. Consistent with the fact that these fungi were (1) isolated from surface-sterilized root fragments[2], (2) enriched in plant roots vs. surrounding soil samples at a continental scale[18] (Fig. 1), and (3) able to recolonize roots of germ-free plants (Supplementary Figs. 10 and 12), both the diversity and the composition of their gene repertoires resemble those of previously described endophytes[28,30,42] (Fig. 2). Unlike the remarkable loss in PCWDE-encoding genes in the genomes of most ectomycorrhizal fungi[31,32], endophytism in root mycobiota members is therefore not associated with genome reduction in saprotrophic traits, as previously suggested[27]. Using a machine learning approach, together with in planta transcriptomic experiments, we identified genes encoding CAZyme families AA9 (copper-dependent lytic polysaccharide monooxygenases, acting on cellulose chains) and GH10 (xylanase) as potential determinants of endophytism (Figs. 3 and 5). Interestingly, these same families were strongly expanded in genomes of beneficial root mutualists belonging to Serendipitaceae[27,35] compared to mycorrhizal mutualists[31] and might therefore represent key genetic components explaining adaptation to—and accommodation in—*A. thaliana* roots. It is important to note that although the 41 isolates are representative of naturally occurring *A. thaliana* root mycobiomes, a large fraction of fungi could not

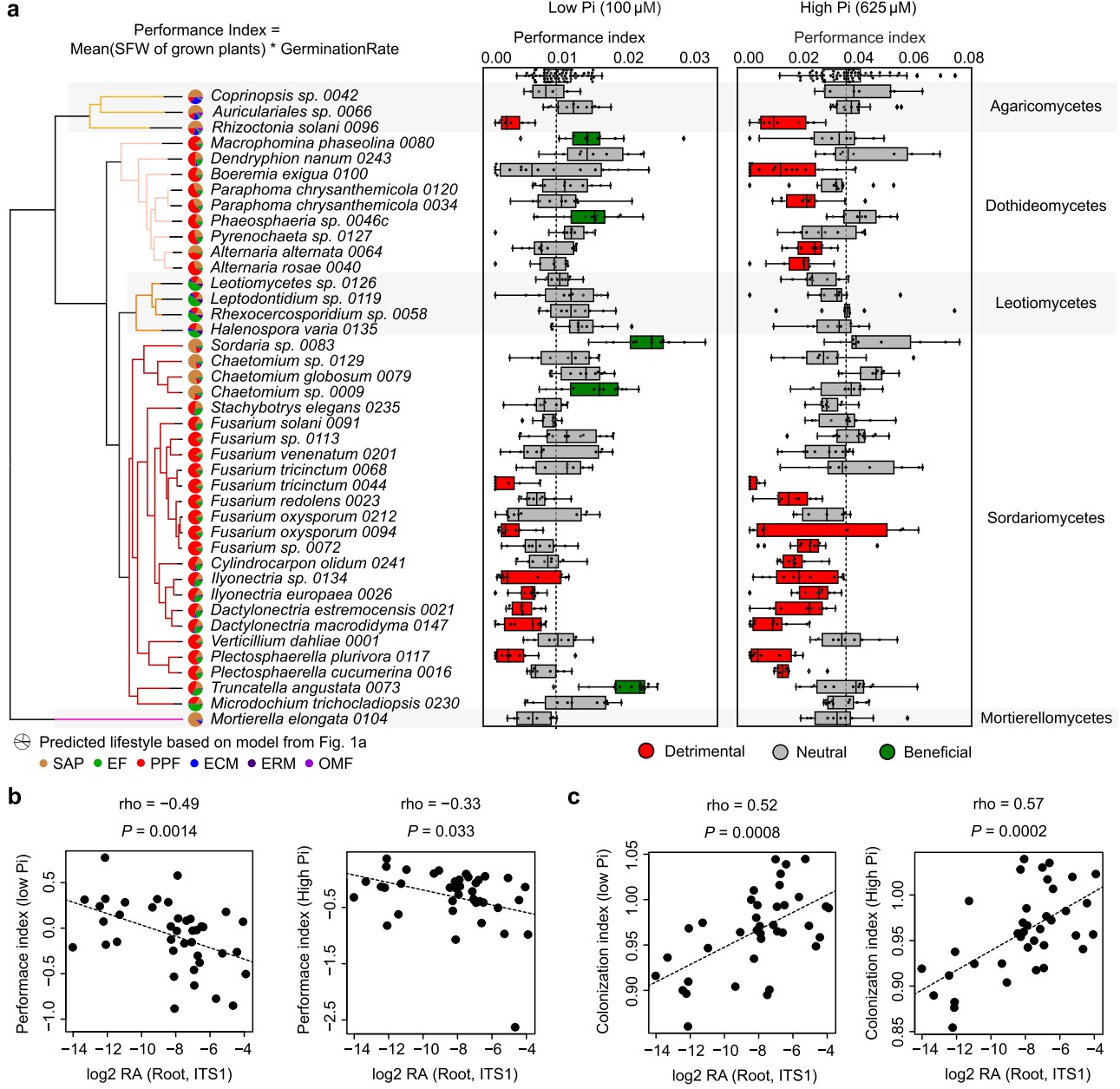

**Fig. 4 Linking fungal outcome on host performance with root colonization patterns. a** Performance indices (shoot fresh weights of 4-week-old plants normalized by germination rate) of *A. thaliana* plants recolonized with each of the 41 fungal strains on media containing low and high concentrations of orthophosphate (Pi). At least three independent biological replicates resulting in 2–4 values each were performed for each fungus ($n = 6$–18). Boxes are delimited by first and third quartiles, central bars show median values, whiskers extend to show the rest of the distribution, but without covering outlier data points (further than 1.5 interquartile range from the quartiles, and marked by lozenges). Differential fungal effects on plant performance were tested on both media with Kruskal–Wallis (at high and low Pi: $P < 2.2e{-}16$) and beneficial and pathogenic strains were identified by a two-sided Dunn test against mock-treated plants (first row in boxplots). Vertical dash lines indicate the mean performance of mock-treated plants. Left to the boxplots is displayed the strain phylogeny, together with lifestyle probabilities predicted by the Random Forest classifier trained for ancestral lifestyle prediction in Fig. 2a. **b** Spearman's rank correlation of relative fungal abundances in root samples from natural populations (log2 RA, see Fig. 1a,[18]) with fungal effects on plant performance at low Pi (left) and high Pi (right) (Hedges standard effect sizes standardizing all phenotypes to the ones of mock-treated plants). **c** Spearman rank correlation of relative fungal abundances in root samples from natural populations (log2 RA, see Fig. 1a,[18]) with fungal colonization indices measured by quantitative PCR in our plant recolonization experiments at low Pi (left) and high Pi (right).

be included in this comparative analysis, including isolates that cannot be cultured. Therefore, it remains to test whether the genomic signatures observed here for this restricted, yet diverse set of cultured fungi, are retained across a broader range of taxonomically diverse root endophytes.

Although the 41 *A. thaliana* root mycobiota members were isolated from roots of healthy-looking plants, experiments in

mono-associations with the host revealed a diversity of effects on plant performance, ranging from highly pathogenic to highly beneficial phenotypes (Fig. 4). These results are consistent with the previous reports[2,24,26,58] and suggest that the pathogenic potential of detrimental fungal endophytes identified based on mono-association experiments with the host, is largely kept at bay in a community context by the combined action of microbiota-

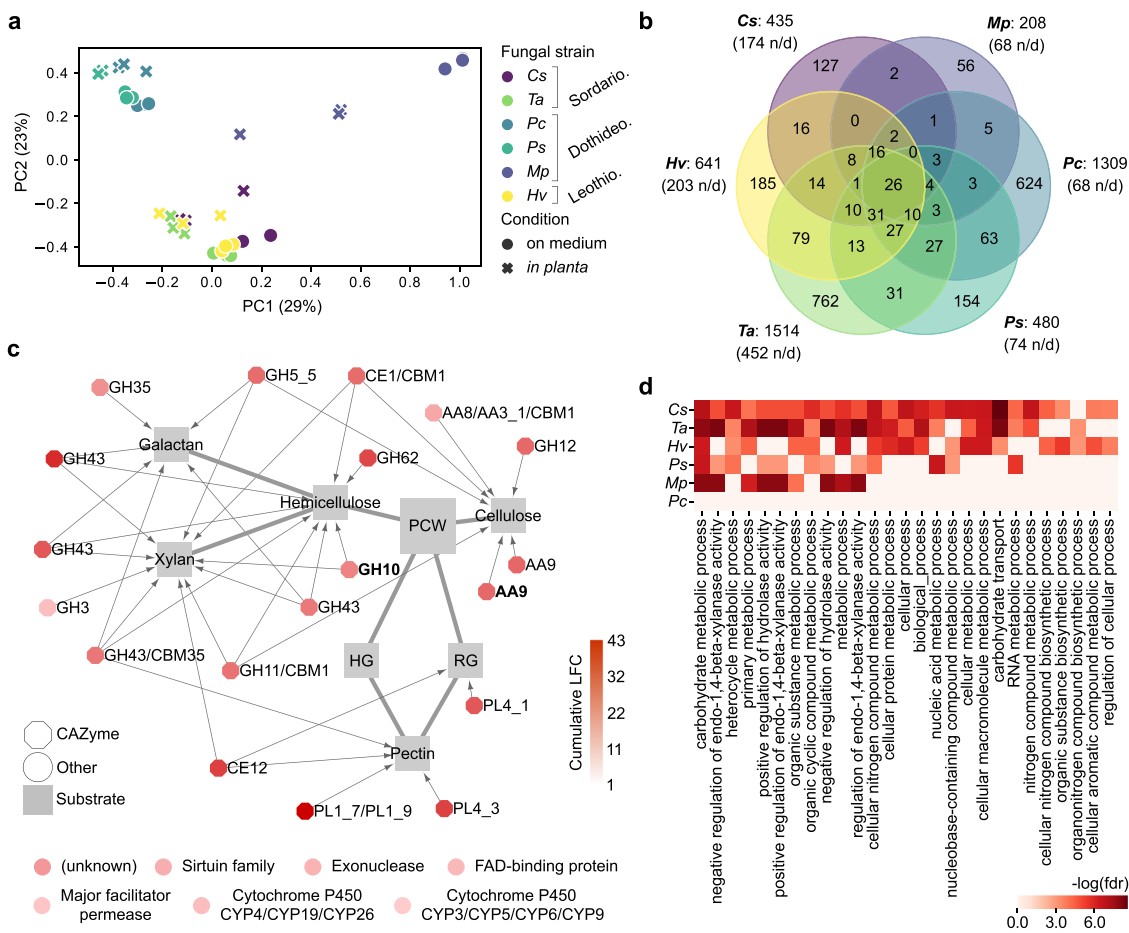

**Fig. 5 Comparative transcriptomics identified a core set of PCWDE-encoding genes induced in *A. thaliana* roots by diverse mycobiota members.**
**a** PCoA plot of Bray-Curtis distances calculated on gene family read counts from fungal transcriptome data on medium and in planta. Cs = *Chaetomium sp. 0009*, Mp = *Macrophomina phaseolina 0080*, Pc = *Paraphoma chrysantemicola 0034*, Ps = *Phaeosphaeria sp. 0046c*, Ta = *Truncatella angustata 0073*, Hv = *Halenospora varia 0135*. **b** Venn diagram showing the number of fungal gene families over-expressed in planta. It highlights 26 families commonly over-expressed by all six fungi (n/d: non-displayed interactions). **c** Commonly over-expressed gene families in planta (n = 26), which include 19 plant cell-wall degrading CAZymes (octagons) linked to their substrates, as described in literature[32,84]. The two CAZyme families highlighted in bold were identified as potential determinants of endophytism (SVM-RFE, see Fig. 3a). The seven remaining (non-CAZyme) families are shown below the network. **d** Individual GO enrichment analyses performed on the genes over-expressed in planta vs. on medium by each fungal strain (GOATOOLS[51], FDR < 0.05).

induced host defenses and microbe-microbe competition at the soil-root interface[2,59–62]. However, we observed that robust and abundant fungal colonizers of *A. thaliana* roots defined from a continental-scale survey of the root microbiota[11] were dominated by detrimental fungi defined based on mono-association experiments with the host (Fig. 4). Based on quantitative PCR data, we also observed that fungi with beneficial activities on plant health were colonizing roots less aggressively than those with detrimental activities—as previously reported[26], suggesting a potential link between fungal colonization capabilities, abundance in natural plant populations, and plant health. A potential limitation of our qPCR-based amplification approach with the general ITS1F-ITS2 primers is linked to the fact that there is copy number variation in rDNA ITS across fungal genomes and that primer bias might distort relative fungal load measurements, thereby making direct comparisons between fungal isolates difficult[63]. Irrespective of this limitation, our results support the idea that maintenance of fungal load in plant roots is critical for plant health, and that controlled fungal accommodation in plant tissues is key for the maintenance of homeostatic plant-fungal relationships. This conclusion is indirectly supported by the fact that an intact innate immune system is needed for the beneficial activities of fungal root endophytes[27,29,62]. Our results, therefore, suggest

that the most beneficial root mycobiota members are not necessarily the most abundant in roots of natural plant populations. In contrast, understanding how potential pathogens can dominate the endospheric microbiome of healthy plants is key for predicting disease emergence in natural plant populations[64,65].

To identify genetic determinants explaining the link between colonization aggressiveness and detrimental effect on plant performance, we used different association methods that all converged into the identification of the CAZyme subfamily PL1_7 as one of the potential underlying determinants of this trait. Proteins from the PL1_7 family were previously characterized in different *Aspergillus* species as metabolizing pectate by eliminative cleavage of (1 -> 4)-α-D-galacturonan[66,67] (EC 4.2.2.2). Furthermore, primary cell walls of *A. thaliana* are enriched with pectin compared to those of monocotyledonous plants, which contain more hemicellulose and phenolics[68,69]. Therefore, repertoire diversity in pectin-degradation capabilities is likely key for penetration and accommodation in pectin-rich *A. thaliana* cell walls. This is corroborated by the observation that non-detrimental fungal endophytes were also shown to consistently induce expression of this gene family in planta during colonization of *A. thaliana* roots (Fig. 5). However, re-inspection of previously published transcriptomic data indicated that genes encoding PL1_7 were

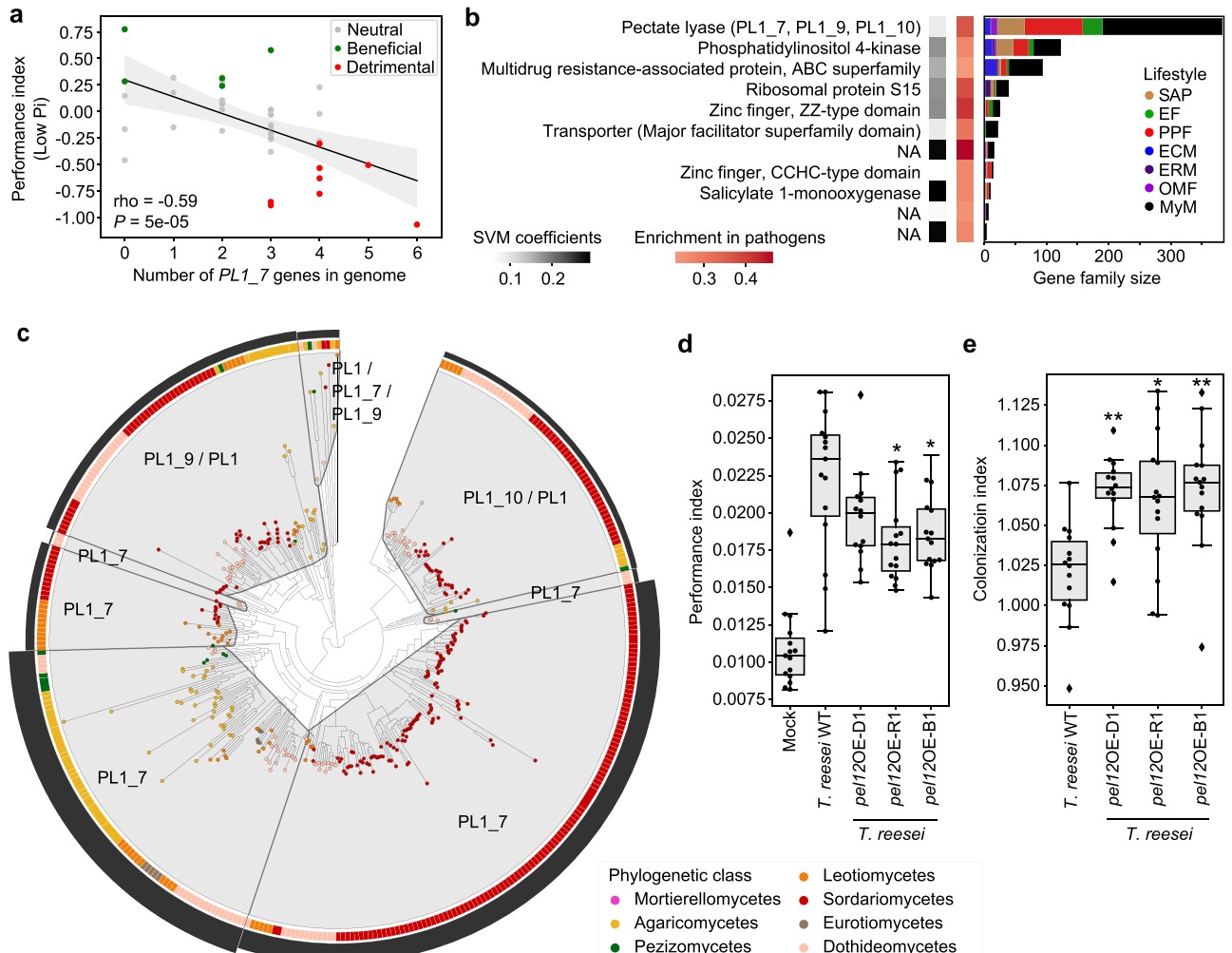

**Fig. 6 Genomic content in polysaccharide lyase PL1_7 links colonization aggressiveness to plant health. a** Spearman's rank correlation between the number of genes encoding secreted PL1_7 in fungal genomes and the plant performance index at low Pi in recolonization experiments. **b** Minimal set of 11 gene families discriminating detrimental from non-detrimental fungi at low Pi (SVM-RFE $R^2 = 0.88$). The first heatmap on the left shows the SVM coefficients, reflecting the contribution of each orthogroup to the separation of the two groups, whereas the heatmap on the right shows the enrichment of these gene families in fungi identified as detrimental in recolonization experiments at low Pi. Gene family sizes and representation in the different lifestyles are shown on the barplots in the context of the whole fungal dataset ($n = 120$). NA: no functional annotation. **c** Protein family tree of the polysaccharide lyase orthogroup identified as essential for segregating detrimental from non-detrimental fungi in our SVM-RFE classification model. The tree was reconciled with fungal phylogeny and clustered into minimum instability groups by MIPhy[56]. Each group is labeled with its CAZyme annotation. The outer circle (black barplot) depicts the relative instabilities of these groups, suggesting two rapidly evolving PL1_7 groups in Sordariomycetes and Agaricomycetes. **d** Plant performance indices resulting from plant recolonization experiments at low Pi (three independent biological replicates), conducted with *Trichoderma reesei* QM9414 (WT) and three independent heterologous mutant lines (D1, R1, B1) overexpressing *pel12* from *Clonostachys rosea* (PL1_7 family,[57]). Asterisks indicate significant difference to *T. reesei* WT, according to ANOVA ($P = 1.45e-12$) and a two-sided TukeyHSD test (WT vs. D1: adjusted $P = 0.28$; WT vs. B1: adjusted $P = 3.75e-2$; WT vs. R1: adjusted $P = 1.19e-2$). **e** Fungal colonization measured by qPCR in colonized roots at low Pi, conducted with *T. reesei* WT and three *pel12* overexpression mutant lines. Asterisks indicate significant difference to *T. reesei* WT, according to Kruskal–Wallis ($P = 6.25e-4$) and a two-sided Dunn test (WT vs. D1: adjusted $P = 2.4e-3$; WT vs. B1: adjusted $P = 1.6e-3$; WT vs. R1: adjusted $P = 1.5e-2$). For both **d** and **e**, three independent biological replicates were performed resulting in $n = 15$ data points per condition. Boxes are delimited by first and third quartiles, central bars show median values, whiskers extend to show the rest of the distribution, but without covering outlier data points (further than 1.5 interquartile range from the quartiles, and marked by lozenges). Asterisks highlight the results of post hoc tests: **adjusted $P < 0.01$, *adjusted $P < 0.05$.

induced more extensively in planta by the fungal root pathogen *Colletotrichum incanum* compared to that of its closely relative beneficial root endophyte *Colletotrichum tofieldiae*[28]. Therefore, differences in expression and diversification of this gene family are potential contributors to the differentiation between detrimental and non-detrimental fungi in the *A. thaliana* root mycobiome, especially since *A. thaliana* cell-wall composition is a determinant factor for disease resistance[70,71]. Notably, expansion

of the PL1_7 gene family was observed in plant pathogens but also in the biocontrol fungus *C. rosea* (Sordariomycetes, Hypocreales), a fungal species with mycoparasitic and plant endophytic capacity[72,73] that is phylogenetically closely related to multiple isolates selected in this study. Genetic manipulation of the *C. rosea pel12* gene revealed a direct involvement of the protein in pectin degradation, but not in *C. rosea* biocontrol towards the phytopathogen *Botrytis cinerea*[57]. Here, we showed that

heterologous overexpression of *C. rosea pel12* in *T. reesei* does not only increase its root colonization capabilities, but also modulates fungal impact on plant performance. We, therefore, conclude that a direct link exists between expression/diversification of PL1_7-encoding genes in fungal genomes, root colonization aggressiveness, and altered plant performance. Our results suggest that the evolution of fungal CAZyme repertoires modulates root mycobiota assemblages and host health in nature.

## Methods

**Selection of 41 representative fungal strains**. The 41 *A. thaliana* root mycobiota members were previously isolated from surface-sterilized root segments of *A. thaliana* and the closely related Brassicaceae species *Arabis alpina* and *Cardamine hirsuta*, as previously described[2]. Notably, this culture collection derived from fungi isolated from the roots of plants grown in the Cologne Agricultural Soil under greenhouse conditions, or from natural *A. thaliana* populations from two sites in Germany (Pulheim and Geyen) and one site in France (Saint-Dié des Vosges)[2] (Supplementary Data 1).

**ITS sequence comparison with naturally occurring root mycobiome**. Comparison of fungal ITS1 and ITS2 sequences with corresponding sequence tags from a European-scale survey of the *A. thaliana* mycobiota (17 European sites[18]) was carried out. For all 41 Fungi, sequences of the internal transcribed spacer 1 and 2 (ITS1/ITS2) were retrieved from genomes (https://github.com/fantin-mesny/Extract-ITS-sequences-from-a-fungal-genome) or, in the cases where no sequences could be found, via Sanger sequencing (4 of 41). All ITS sequence variants were directly aligned to the demultiplexed and quality filtered reads from previously published datasets[18] using USEARCH[74] v10.0.240 at a 97% similarity cut-off. A count table across all samples was constructed using the results from this mapping and an additional row representing all the reads that did not match any of the reference sequences was added. This additional row was based on the count data from the amplicon sequence variant (ASV) analysis from the original study, whereas the read counts from the new mapping were subtracted sample wise. To have coverage-independent information on the RA of each fungus, we calculated RA only for the root samples where the respective fungi were found (RA > 0.01%). The sample coverage was calculated across all root samples (>1000 reads, $n = 169$). Enrichment in roots was calculated for all root and soil samples (>1000 reads, $n = 169 / n = 223$) using the Mann–Whitney *U* test (FDR < 0.05). In the same way the RA and coverage across leaf samples from two *A. thaliana* populations[39] was calculated (two locations in Germany, samples $n = 51$). For this specific analysis of leaf samples, only ITS2 sequences were used and no fold change was calculated. In order to estimate the presence of the 41 fungi across worldwide collected samples, we used the GlobalFungi database[38] (https://globalfungi.com/, version August 2020). The most prevalent ITS1 sequences from each genome were used to conduct a BLAST search on the website. Sample metadata for the best matching representative species hypothesis sequences were then used to determine the global sample coverage. Appearance across samples from type root/shoot was counted for each fungus and compared to the total number of root/shoot samples for each continent.

**Whole-genome sequencing and annotation**. Forty-one fungal isolates from a previously assembled culture collection[2] were revived from 30% glycerol stocks stored at −80 °C. Genomic DNA extractions were carried out from mycelium samples grown on Potato extract Glucose Agar (PGA) medium, with a previously described modified cetyltrimethylammonium bromide protocol[31]. Genomic DNA was sequenced using PacBio systems. Genomic DNA was sheared to 3 kb, > 10 kb, or 30 kb using Covaris LE220 or g-Tubes or Megaruptor3 (Diagenode). The sheared DNA was treated with exonuclease to remove single-stranded ends and DNA damage repair mix followed by end repair and ligation of blunt adapters using SMRTbell Template Prep Kit 1.0 (Pacific Biosciences). The library was purified with AMPure PB beads and size selected with BluePippin (Sage Science) at >10 kb cutoff size. Sequencing was done on PacBio RSII or SEQUEL machines. For RSII sequencing, PacBio Sequencing primer was annealed to the SMRTbell template library and sequencing polymerase was bound to them. The prepared SMRTbell template libraries were sequenced on a Pacific Biosciences RSII or Sequel sequencers using Version C4 or Version 2.1 chemistry and 1 × 240 or 1 × 600 movie run times, respectively. The genome assembly was generated using Falcon[75] v0.7.3 with mitochondria-filtered reads. The resulting assembly was improved with finisherSC, and polished with either Quiver or Arrow. Transcriptomes were sequenced using Illumina Truseq Stranded RNA protocols with polyA selection (http://support.illumina.com/sequencing/sequencing_kits/truseq_stranded_mrna_ht_sample_prep_kit.html) on HiSeq2500 using HiSeq TruSeq SBS sequencing kits v4 or NovaSeq6000 using NovaSeq XP v1 reagent kits, S4 flow cell, following a 2 × 150 indexed run recipe. After sequencing, the raw fastq file reads were filtered and trimmed for quality (Q6), artifacts, spike-in, and PhiX reads and assembled into consensus sequences using Trinity[76] v2.1.1.

The genomes were annotated using the JGI Annotation pipeline[77]. Species assignment was conducted by extracting ITS1 and ITS2 sequences from genome assemblies, performing a similarity search against the UNITE database[78] (https://unite.ut.ee, version February 2021) and a phylogenetic comparison to fungal genomes on MycoCosm[77] (https://mycocosm.jgi.doe.gov).

**Comparative genomics dataset**. In addition to our 41 fungal isolates from *A. thaliana* roots, we used 79 previously published fungal genomes in a comparative genomics analysis (Supplementary Data 2). While 77 genomes and annotations were downloaded from MycoCosm, the genome assemblies of fungal strains *Harpophora oryzae* R5-6-1[34] and *Helotiales sp.* F229[30] were downloaded from NCBI (GenBank assembly accessions GCA_000733355.1 and GCA_002554605.1 respectively) and annotated with FGENESH[79] v8.8.0. Lifestyles were associated to each single strain by referring to the original publications describing their isolation, and consulting the FunGuild[40] database with the species and genus names associated to each strain. Orthology prediction was performed on this dataset of 120 genomes by running OrthoFinder[46] v2.2.7 with default parameters. From this prediction, we used the generated orthogroups data, the species tree, and gene trees. OrthoFinder was also run on our 41 newly sequenced fungi to obtain a second species tree, for this subset.

**Predicting ancestral lifestyles**. To identify gene family gains and losses events, GLOOME[80] *gainLoss.VR01.266* was run using the species tree and presence/absence of each orthogroup in the 120 genomes. To obtain reconstruction of ancestral genomes using the Wagner parsimony approach, Count[47] v10.04 was run using these same inputs. To associate a lifestyle to each reconstructed ancestral genome, a Random Forest classifier was trained on the copy numbers of each orthogroup in the comparative genomics dataset excluding *A. thaliana* mycobiota members, and the fungal lifestyles associated to these 79 genomes. This was performed using the RandomForestClassifier() function of the Python library *sklearn*[81] v0.20.3. The accuracy of the model was estimated by a leave-one-out cross-validation approach, computed using the function cross_val_score(cv=KFold(n_splits=120)) in *sklearn*. Finally, the probabilities of ancestors to belong in each lifestyle category were retrieved using function predict_proba().

**Genomic feature analyses**. Statistics of genome assemblies (i.e., N50, number of genes and scaffolds and genome size) were obtained from JGI MycoCosm[77], and assembly-stats (https://github.com/sanger-pathogens/assembly-stats). Genome completeness with single copy orthologues was calculated using BUSCO v3.0.2 with default parameters[82]. The coverage of transposable elements in genomes was calculated and visualized using a custom pipeline Transposon Identification Nominative Genome Overview (TINGO[83]). The secretome was predicted as described previously[49]. We calculated, visualized, and compared the count and ratio of total (present in the genomes) and predicted secreted CAZymes[84], proteases[85], lipases[86], and small secreted proteins[49] (SSPs) (<300 amino acid) as a subcategory. We calculated the total count of the followings using total and predicted secreted plant cell-wall degrading enzymes (PCWDEs) and fungal cell-wall degrading enzymes (FCWDEs). Output files generated above were combined and visualized with a custom pipeline, Proteomic Information Navigated Genomic Outlook (PRINGO[32]). To compare the genomic compositions of the different lifestyle categories while taking into account phylogenetic signal, we first generated a matrix of pairwise phylogenetic distances between genomes (*i.e.* sum of branch lengths) using the function tree.distance() from package *biopython Phylo*[87], then computed a principal component analysis using the PCA(n_components=4) function of *sklearn*[81] v0.20.3. Components PC1, PC2, PC3 and PC4 (Supplementary Fig. 3) were then used to compare the per-genome numbers of CAZymes, proteases, lipases, SSPs, PCWDEs, and FCWDEs in the different lifestyles with an ANOVA test and a TukeyHSD post hoc test. R function aov() was used with the following formula specifying the model:

$$GeneCount \sim PC1 + PC2 + PC3 + PC4 + Lifestyle$$

$$+ PC1 : Lifestyle + PC2 : Lifestyle + PC3 : Lifestyle + PC4 : Lifestyle$$

Differences in subfamily composition of the groups of genes of interest were then carried out using a PERMANOVA-based approach (https://github.com/fantin-mesny/Effect-Of-Biological-Categories-On-Genomes-Composition). This approach relies on Jaccard distances calculation (best suited for discrete variables such as copy numbers) then a PERMANOVA testing with function adonis2() from R package *Vegan* v2.5-7 (https://github.com/jarioksa/vegan), with the model specified by the following formula:

$$JaccardDistanceMatrix \sim PC1 + PC2 + PC3 + PC4 + Lifestyle$$

$$+ PC1 : Lifestyle + PC2 : Lifestyle + PC3 : Lifestyle + PC4 : Lifestyle$$

Post hoc testing with function pairwise.perm.manova() from package *RVAide-Memoire* v0.9-77 (https://cran.r-project.org/web/packages/RVAideMemoire) was then performed to compare pairs of lifestyles.

For each Jaccard matrix, we used the function dbRDA() from the R package *Vegan*, to calculate two distance-based redundancy analyses (dbRDA), respectively constrained by phylogenetic variables (formula *Jaccard~Condition(Lifestyle)+PCs*) and by lifestyle groups (formula *Jaccard~Condition(PCs)+Lifestyle*).

We determined genes discriminating groups based on the principal coordinates of a regularized discriminant analysis calculated from the count of genes coding for CAZymes, proteases, lipases, and SSPs, with R function rda(). We then used *Vegan* function scores() on the three first principal coordinates, and kept for each coordinate the top five high-loading gene discriminating groups.

**Determinants of endophytism**. To identify a small set of orthogroups that best segregate endophytes and mycobiota members from fungi with other lifestyles, we standardized the orthogroup gene counts with function StandardScaler() from *sklearn*[81] v0.20.3. Then, orthogroups that are enriched or depleted in the fungi of interest were selected with function SelectFdr(f_classif, alpha=0.05) from *sklearn*. On this subset of orthogroups, we trained a Support Vector Machine classifier with Recursive Feature Elimination (SVM-RFE). This was performed with functions from *sklearn* SVC(kernel = 'linear') and RFECV(step=10, cv=KFold(n_s-plits=120, min_features_to_select=10)), which implement a leave-one-out cross-validation allowing the estimation of the classifier accuracy at each step of the recursive orthogroup elimination. PhyloGLM models[50] were built with R package *phylolm* v.2.6.2 on the two groups of interest and orthogroup gene counts, with parameters btol = 45 and log.alpha.bound = 7, and the *logistic_MPLE* method. Further analysis of the gene families segregating fungi of interest from others ($n = 84$) was carried out by identifying a representative sequence of each orthogroup in our SVM-RFE model, and studying both its annotation and coexpression data in databases. To identify representative sequences, all protein sequences composing an orthogroup were aligned with FAMSA[88] v1.6.1. Using HMMER[89] v3.2.1, we then built a Hidden Markov Model (HMM) from this alignment with function hmmbuild, then ran function hmmsearch looking for the best hit matching this HMM within the proteins composing our orthogroup. We then considered this best hit as a representative sequence of the orthogroup and analyzed its annotation. GO enrichment analysis was performed by running GOATOOLS[51] v1.0.3 using the GO annotations associated to the representative sequences. To obtain coexpression data linking the orthogroups retained in our SVM-RFE model, we searched the String-db[52] website (https://string-db.org, version August 2020) for COG protein families matching our set of representative protein sequences in fungi. Each protein was associated to one COG (Supplementary Data 4a), and coexpression data were downloaded. A coexpression network was then built on the families enriched in endophytes and mycobiota members ($n = 73$) and clustered with algorithm MCL (granularity = 5) using Cytoscape[90] v3.7.2 and clusterMaker2[91] v1.3.1.

**Plant recolonization experiments assessing the effect of each fungal strain on plant growth**. *A. thaliana* seeds were sterilized 15 min in 70% ethanol, then 5 min in 8% sodium hypochlorite. After six washes in sterile double-distilled water and one wash in 10 mM MgCl₂, they were stratified 5–7 days at 4 °C in the dark. Seed inoculation with fungal strains was carried out by crushing 50 mg of mycelium grown for 10 days on Potato extract Glucose Agar medium (PGA) in 1 ml of 10 mM MgCl₂ with two metal beads in a tissue lyser, then adding 10 µM of this inoculum in 250 µl of seed solution for 5 min. Seeds were then washed twice with MgCl₂ before seven were deposited on each medium-filled square Petri plate. Mock-inoculated seeds were also prepared by simple washes in MgCl₂. The two media used in this study—625 and 100 µM Pi—were previously described[92]. They were prepared by mixing 750 µM MgSO₄, 625 µM/100 µM KH₂PO₄, 10,300 µM NH₄NO₃, 9400 µM KNO₃, 1500 µM CaCl₂, 0.055 µM CoCl₂, 0.053 µM CuCl₂, 50 µM H₃BO₃, 2.5 µM KI, 50 µM MnCl₂, 0.52 µM Na₂MoO₄, 15 µM ZnCl₂, 75 µM Na-Fe-EDTA, and 1000 µM MES pH5.5, 0 µM/525 µM KCl, then adding Difco™ Agar (ref. 214530, 1% final concentration), and finally adapting the pH to 5.5 prior to autoclaving. Plants were grown for 28 days at 21 °C, for 10 h with light (intensity 4) at 19 °C and 14 h in the dark in growth chambers. While roots were harvested and flash-frozen, SFW was measured for each plant. To distinguish seeds that did not germinate from plants that could not develop because of a fungal effect, we introduced a per-plate PPI corresponding to the average SFW of grown plants multiplied by the proportion of grown plants. In further correlation analyses, we used plant-performance indexes normalized to mock controls (standard effect sizes) using the Hedges' *g* method[93].

**Fungal colonization of roots assay**. Frozen root samples (one per plate) were crushed and total DNA was extracted from them using a QIAGEN Plant DNEasy Kit. Fungal colonization of these root samples was then measured by quantitative PCR. For each sample, two reactions were conducted with primers ITS1F (5′-CTTGGTCATTTAGAGGAAGTAA-3′) and ITS2 (5′-GCTGCGTTCTTCATC-GATGC-3′) which target the fungal ITS1 sequence, and two with primers UBQ10F (5′-TGTTTCCGTTCCTGTTATCT-3′) and UBQ10R (5′-ATGTTCAAGC-CATCCTTAGA-3′) that target the *Ubiquitin10 A. thaliana* gene. Each reaction was performed by mixing 5 µl of iQ™ SYBR® Green Supermix with 2 µl of 10 µM forward primer, 2 µl of 10 µM reverse primer and 1 µl of water containing 1 ng template DNA. A BioRad CFX Connect Real-Time system was used with the following programme: 3 min of denaturation at 95 °C, followed by 39 cycles of 15 sec at 95 °C, 30 s at 60 °C and 30 s at 72 °C. We then calculated a single colonization index for each sample using the following formula: Index = $2^{-Cq(ITS1)/Cq(UBQ10)}$.

**Confocal microscopy of root colonization by fungi**. Roots of plants grown for 28 days in mono-association with fungi were harvested and conserved in 70% ethanol. They were then rinsed in ddH2O, and stained with propidium iodide (PI) and wheat germ agglutinin conjugated to fluorophore Biotium CF®488 (WGA-CF488). This was carried out by dipping the root samples for 15 min in a solution of 20 µg/ml PI and 10 µg/ml WGA-CF488 buffered at pH 7.4 in phosphate-buffered saline (PBS). Samples were then washed in PBS and imaged with a Zeiss LSM700 microscope and the associated software ZEN v2.3 SP1.

**Plant-fungi interaction transcriptomics**. Dual RNAseq of six different plant-fungi interactions was carried out by performing three independent plant recolonization experiments on our low Pi medium, as described above. Total roots per plates were harvested after 28 days in culture, flash frozen, and crushed in a tissue lyser, and then total RNA was extracted with a QIAGEN RNeasy Plant Mini kit. As a control condition, sterile Nucleopore Track-Etched polyester membranes were deposited on low Pi medium, then 10 µl drops of fungal inoculum (50 mg/ml of mycelium in 10 mM MgCl₂) were placed on each one. The membranes were collected and processed as the root samples of our test condition. PolyA-enrichment was carried out on the RNA extracts, then an RNAseq library was prepared with the NEBNext Ultra™ II Directional RNA Library Prep Kit for Illumina (New England Biolabs). Sequencing was then performed in single read mode on a HiSeq 3000 system. RNAseq reads were trimmed using Trimmomatic[94] v0.38 and parameters TRAILING:20 AVGQUAL:20 HEADCROP:10 MINLEN:100. We then used HiSat2[54] v2.2.0 to map the trimmed reads onto reference genomes. Six independent HiSat2 indexes were prepared, each based on the TAIR10 *A. thaliana* genome and one of the six fungal genome assemblies of interest. We then performed six map-pings, and counted the mapped reads using featureCounts[95] v2.0.0. RPKM (Reads Per Kilobase Million) values were computed from the featureCounts output. Differential gene expression analyses were then carried out on these counts with DESeq2[55] v1.24.0. log2FC values were corrected by shrinkage with the algorithm *apeglm*[96] v1.6.0. To compare the transcriptomes of the six different fungi, significant log2FC values were summed per orthogroup. For each orthogroup, we used annotation of the most representative sequence, as previously described. GO enrichment analyses were carried out with GOATOOLS[51] v1.0.3, using the MycoCosm[77] GO annotation for fungi, and the TAIR annotation for *A. thaliana*.

**Determinants of detrimental effects on plants and analysis of pectate lyases**. Determinants of detrimental effects at low Pi were identified with the same method as previously described for determinants of endophytism/mycobiota: standard scaling of the orthogroup gene counts, then training an SVM classifier with RFE and leave-one-out cross validation. Instability analysis was carried out by submitting the species tree generated by OrthoFinder[46] to MIPhy[56] (http://miphy.wasmuthlab.org, version October 2020), together with the gene tree of our orthogroup of interest, with default parameters. PhyloGLM[50] models were built with R library *phylolm* v.2.6.2 on the two groups detrimental/non-detrimental and CAZyme gene counts, using our 41-genome species tree with default parameters and the *logistic_MPLE* method. *T. reesei* strain QM9414 and three heterologous overexpression lines of *pel12* generated previously[57], were revived on PGA medium and then inoculated into seeds for plant recolonization experiments on low Pi medium as previously described.

**Statistics**. Except for statistical methods described in the previous paragraphs, statistical testing was performed in R v3.5.1. Function aov() was used for ANOVA tests. Two-sided TukeyHSD post hoc testing was performed using function TukeyHSD(), which compares values associated to the different categories of one factor, respective of the variance that was attributed to this factor by the previous ANOVA test. When data were abnormally distributed, the non-parametric Kruskal–Wallis test was used by running function kruskal.test(), and the two-sided Dunn post hoc test was performed with function DunnTest() from package *DescTools* v0.99.28 (https://github.com/AndriSignorell/DescTools/).

**Reporting summary**. Further information on research design is available in the Nature Research Reporting Summary linked to this article.

## Data availability

The genomic data generated in this study have been deposited in the GenBank database under the following BioProject accession codes: PRJNA371205 (assembly JAHBNJ000000000), PRJNA347188 (assembly JAHBNI000000000), PRJNA441695 (assembly JAHBNH000000000), PRJNA370201 (assembly JAGJXA000000000), PRJNA571620 (assembly JAGIZQ000000000), PRJNA370120 (assembly JAHBOE000000000), PRJNA347200 (assembly JAHBOF000000000), PRJNA371203 (assembly JAGPYM000000000), PRJNA370196 (assembly JAGMUU000000000), PRJNA500112 (assembly JAGMUV000000000), PRJNA370194 (assembly JAGMWT000000000), PRJNA455444 (assembly JAHBOG000000000), PRJNA370199 (assembly JAHBOO000000000), PRJNA347190 (assembly JAHEWL000000000), PRJNA455442 (assembly JAHEVI000000000), PRJNA347185 (assembly JAGMUX000000000), PRJNA370198 (assembly JAGTJS000000000), PRJNA347189 (assembly JAGPXF000000000), PRJNA455443 (assembly JAGMVH000000000),

PRJNA500113 (assembly JAGMVI000000000), PRJNA347186 (assembly JAGPNQ000000000), PRJNA347191 (assembly JAHLEZ000000000), PRJNA370195 (assembly JAGTJR000000000), PRJNA370119 (assembly JAGTJQ000000000), PRJNA347187 (assembly JAGSXK000000000), PRJNA500111 (assembly JAHEWK000000000), PRJNA347192 (assembly JAGTJP000000000), PRJNA347193 (assembly JAGMWG000000000), PRJNA538399 (assembly JAGMVJ000000000), PRJNA459235 (assembly JAGTJN000000000), PRJNA347194 (assembly JAGMVK000000000), PRJNA371204 (assembly JAGPXD000000000), PRJNA570880 (assembly JAGSXJ000000000), PRJNA347196 (assembly JAGTJM000000000), PRJNA347195 (assembly JAGTJL000000000), PRJNA371202 (assembly JAGTJO000000000), PRJNA370118 (assembly JAGPNK000000000), PRJNA370200 (assembly JAGPNJ000000000), PRJNA347197 (assembly JAGPXC000000000), PRJNA519173 (assembly JAHEWJ000000000), and PRJNA370197 (assembly JAHEWH000000000). The transcriptomic data generated in this study have been deposited in the Gene Expression Omnibus database under accession code GSE169629. The processed transcriptomic data are also available in this GEO entry. We referred to three online databases for analysis: UNITE (https://unite.ut.ee, version February 2021), GlobalFungi (https://globalfungi.com, version August 2020) and String-db (https://globalfungi.com/, version August 2020). The plant phenotypic data and fungal colonization values are provided in the Source Data file. Source data are provided with this paper.

## Code availability

All the scripts used for data processing and analysis were written in Python v3.7.3 and R v3.5.1 (except for transcriptomic analyses in which R v3.6.1 was used). Scripts are available at GitHub (https://github.com/fantin-mesny/Scripts-from-Mesny-et-al.-2021)[97].

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

## Acknowledgements

The sequencing project was funded by the U.S. Department of Energy (DOE) Joint Genome Institute, a DOE Office of Science User Facility, and supported by the Office of Science of the U.S. DOE under Contract No. DE-AC02-05CH11231 within the framework of CSP 1974 "1KFG: Deep-sequencing of ecologically relevant Dikaria". This work was supported by funds to S.Hac from a European Research Council starting grant (MICRORULES 758003), the 'Priority Programme: Deconstruction and Reconstruction of the Plant Microbiota (SPP DECRyPT 2125)' and the Cluster of Excellence on Plant Sciences (CEPLAS), both funded by the Deutsche Forschungsgemeinschaft. F.M. salary was covered by the DECRyPT 2125 programme. This research was also supported by the Laboratory of Excellence ARBRE (ANR- 11-LABX-0002-01), the Region Lorraine, the European Regional Development Fund, and the Plant–Microbe Interfaces Scientific Focus Area in the Genomic Science Program, the Office of Biological and Environmental Research in the US DOE Office of Science (to F.M.M.). M.K. acknowledges funding from the SLU Centre for Biological Control (CBC). The Austrian Science Fund FWF project P30460-B32 is acknowledged for funding L.A. We would like to thank Nathan Vannier for regular discussions and ideas about data analysis and method development. Finally, we thank Paul Schulze-Lefert, Ruben Garrido-Oter, Ryohei Thomas Nakano, Gregor Langen and Rozina Kardakaris for providing helpful comments regarding the manuscript or during departmental seminars and thesis advisory committee meetings.

## Author contributions

S. Hacquard and F.M.M. initiated, coordinated and supervised the project. Genomic analyses were performed by F.M., with help of S.M. who provided scripts and pipelines to annotate genomic features and generate figures describing the structure and composition of fungal genomes. Transcriptomic analyses were performed by F.M. T.T. re-analyzed ITS amplicon sequencing data from natural site samples. F.M. performed all the experiments, with technical assistance from B.P. L.A. and M.K. provided fungal mutant strains used for functional validation of PL1_7. K.W.B., S. Haridas, C.C., D.B., A.L., W.A., J.P., K.L., R.R., A.C., and I.V.G. sequenced, assembled and annotated the fungal genomes. E.D. and B. Henrissat annotated CAZymes. B. Hüttel. prepared RNAseq libraries and sequenced the transcriptomes used for differential gene expression analysis. F.M. and S. Hacquard wrote the manuscript, with input from S.M., A.K., and F.M.M.

## Funding

## Competing interests

The authors declare no competing interests.
