## [Peer Review File · Nature Communications]

Genetic determinants of endophytism in the *Arabidopsis* root mycobiomeREVIEWER COMMENTS

Reviewer #1 (Remarks to the Author):

This is a very thorough work and a well written paper.

I have only rather minor comments, and mostly regarding presentation and interpretation
1 the general assumption seems to be that (majority of) root-colonising fungi are culturable. To which degree this holds for endophytes? Clearly, for mycorrhizal fungi this is not true. The paper provides information on how the field-collected abundance and experimentally inoculated abundance of the focal fungal isolates (well and positively) correlate. But is there any caveats here, an aspect missed if non-culturable fungi are not considered?

2 the mycorrhizal fungi addressed in some analyses (genomics) include ectomycorrhiza, ericoid mycorrhiza and orchid mycorrhiza, but not arbuscular mycorrhiza. Why? Genomes of number of AMF are available. At minimum, it should be acknowledged when interpreting the data that AMF may be quite a different lot, genomically and genetically speaking

L47: rich diversit - do you mean high diversity? Or rich set of?

L176: the plant performance index combining both plant biomass and germination rate is very nice idea. How do the two components contribute to the general index, how variable are they? Any fungi influencing more the germination vs growth?

Fig 4: there is large variation in the effect of some fungi on plant performance. Is this a notable result? This is, fungi which effect is hard to predict, or potentially very context dependent (and hence, potentially could be managed/directed towards a desired outcome)?

Fig 4 a: what are the dashed lines indicating?

Reviewer #2 (Remarks to the Author):

The manuscript by Mesny et al. provides a thorough assessment of the evolution of endophytism in fungi associated with roots of *Arabidopsis thaliana*, and the genomic signatures resulting from it. Focusing on a core set of root mycobiome members of that plant species, the authors re-analyzed existing amplicon sequencing data to evaluate their prevalence in roots across a wide geographic range, performed re-colonization experiments to test the effects of fungi on host growth, and did comparative genomics to identify genetic determinants of the endophytic lifestyle. The latter resulted in a list of candidate gene families best explaining the endophytic lifestyle, one of which was validated in recolonization experiments with existing mutants of a fungus over-expressing one target gene.

Overall, I think this is a comprehensive study that affords interesting insights into the functional ecology and plant interactions of a fungal group that, despite making up a major component of fungal communities in plants, have poorly understood associations with plants. Besides, it provides a very valuable dataset in the form of genome data from a fair number of fungi that are common root endophytes, not only of *Arabidopsis thaliana* but also from other plant hosts and habitats. Indeed, the main strength of the study lies in the amount of genomic data generated and the inferences made from them, whereas the data concerning the prevalence of fungi in roots and the effects on plant growth confirm results from several previous studies. Several such studies are properly cited in the manuscript.

I only have a few comments about the manuscript, the first one dealing with its structure. Although the manuscript is well and clearly written, it does not provide upfront an specific explanation of the study background, aims, and hypotheses. Instead, the 'storyline' develops as the paper advances, so that specific questions are posited in the Results section just before being tackled. The Introduction is in fact very short and reads more like an extended abstract, with only a general exposé of the antecedents at the beginning (L47-56) followed by a summary of the research done (L56-64). Besides, specific aspects important for the research are not even introduced, as if the authors relied on the reader's knowledge of the relevant literature. For example, it is not explicitly explained why transposable elements (L125-126) were investigated, or why do some genes (L128-130) are expected to play important roles in fungal-host interactions rather than being common for, e.g., saprotrophy. Altogether, the way it is presented, the

manuscript reads like an explorative study in which the authors did a lot of work and then looked for patterns in the results. I suggest the authors to clearly state upfront their motivation for doing this research, explicitly mentioning their specific aims and hypotheses or expectations, as is commonly requested for most scientific papers.

Many of the methodological approaches used are above my paygrade and thus I cannot judge their specifics, although in general they are explained well enough to understand what was done. I do wonder about the ability to properly disentangle the effects of fungal lifestyles from those of phylogeny. Even though the authors selected quite a comprehensive list of fungal strains representing different life strategies (please, note that shoe-horning fungi into unique categories is often difficult; e.g. see Selosse et al. 2018. *New Phytologist* 217: 968–972.) and used different methods to account for phylogenetic signal (e.g. introducing phylogeny PCs in ANOVA tests or using PhyloGLM models), the phylogenetic representation of lifestyles is rather biased: e.g. with many closely related pathogenic members of the Nectriaceae, endophytes in the Helotiales, EcM in the Agaricales, and only a couple of related OcM (Fig. S2). I acknowledge that the authors' efforts are probably the best that can be done, given the strong phylogenetic conservatism in some lifestyles. But my concerns lie on the capacity to statistically account for that conservatism. For example, do the letters above boxplots in Fig. 2b represent differences between lifestyles once phylogeny has been accounted for, or they represent instead differences by both phylogeny and lifestyle (this is not clear from the legend)? How would plots in Fig. S4 look if they were to show only the variance explained by lifestyle (e.g. 'vegan::dbrda(JaccardMatrix ~ Condition(Phylogeny) + Lifestyle)')? At any rate, in the relevant parts of the manuscript the authors should specify which amount of data variation is exclusively explained by lifestyle and phylogeny, and which amount cannot be differentiated between both factors (e.g. variation partitioning).

My last comment refers to the relationship found between fungal abundance in roots and the detrimental effects on plant growth. This relationship makes sense and has indeed been shown in the past (see Kia et al. 2017. *The ISME Journal* 11: 777–790). However, here, the quantification of fungi in roots relied on qPCR with general fungal primers that target the rDNA ITS, a region with variable copy numbers across fungi (Lofgren et al. 2019. *Molecular Ecology* 28: 721–730) that would therefore distort direct comparisons across species (other fungal traits, such as nuclei per cell, cells per hyphal length, etc. might also influence comparisons across species). Note also that, at least in bacteria, rDNA copy numbers are correlated with different life strategies (Klappenbach et al. 2000. *Applied and Environmental Microbiology* 66: 1328–1333.) or growth habits that could also affect, per se, interactions (again, see Kia et al. 2017). Some discussion about the validity and/or the caveats of the assessment should be included.

Specific comments:

L75: r_s -> please replace by ' ρ ', the standard representation of the Spearman's correlation coefficient (also in Fig. 1b and elsewhere in the text).

L76: $n = 30/41$ -> does this refer to the number of samples, out of the total, in which RA was above 0.1%. Please, state clearly. Same for L77.

L80-82: I have problems understanding the two panels in Fig. 1c. Please, describe more clearly.

L112: what are the grey sectors in the pie charts? Do they correspond to MyM? If that's the case, I wonder whether this category is comparable to the predefined lifestyles (SAP, EF, etc.): the latter summarize functional strategies of interaction with plant hosts, whereas MyM refers to the origin of the isolates (i.e. from roots of Arabidopsis). My point here is: do large fractions of gray in ancestral nodes indicate that ancestors were likely 'Arabidopsis mycobiota members'? The figure may be more clear if 'MyM' is replaced by 'unknown'.

L112-115: the two examples here are not evident in Fig. 2a. Perhaps you could highlight in the figure the specific tree tips referred to in the main text.

L133: do letters above boxplots in Fig. 2b indicate differences exclusively based on lifestyle, or on both lifestyle and phylogeny? Please, indicate this explicitly, either here or in the figure legend (L828). I'd say that only differences purely based on lifestyle are relevant here.

L297-300: please note that a similar relationship between root colonization aggressiveness and detrimental effects on host growth has been previously described (Kia. et al. ISME J 11, 777–790 (2017)).

L176: 'of five-week-old plants' -> '4-week-old plants' in legend of Fig. 4 (L887), and L484. Please, reconcile.

L430: wouldn't it be better to use PCoA than PCA to obtain the phylogenetic PCs? The former performs better in representing non-euclidean distances, such as phylogenetic distances (e.g. Desdevises et al. 2003. Evolution 57: 2647–2652.). In addition, why only select the first two PCs? These might only collect the phylogenetic separation of a few major lineages, whereas additional PCs might provide a finer-grained representation of the phylogenetic relationships among fungi (a common approach involves variable selection for selecting relevant PCs). In any case, please provide the PCA/PCoA with the proportion of variance explained by the selected PCs.

L787: ...but based on...

L948: 'The first vertical stripe' -> The vertical stripe to the right?

L949: 'whereas the second' -> whereas the one to the left?

L950: 'Gene family sizes and representation in the different lifestyles...' -> but is this an absolute measure? Wouldn't it be biased by the number of fungi included in each dataset? If that's the case, please normalize by genome (e.g. median size?)

L1211, 1213: 'sp.' not in italics

L1253, 1256: 'On the right' -> below?

Reviewer #3 (Remarks to the Author):

Summary: Mesney et al., sequenced the genomes of 41 fungal endophytes previously isolated from Arabidopsis roots with the goal to identify a genomic basis for root endophytism. Arabidopsis endophytes were compared to 79 previously sequenced fungal genomes representing phylogenetically and ecologically diverse taxa. Comparative genomic analyses and machine learning methods identified a subset of genes that appear to be enriched in root symbionts and other endophytes. The authors also found that root endophyte impact on host plant performance varied from detrimental to beneficial, which they linked to the degree of root colonization. Fungi that were able to colonize more root tissue had expanded pectin gene repertoires and were more likely to be detrimental to host health.

Overall, I felt that the manuscript was well written, and the rationale, results, and implications of the work were clear. I believe the authors major claims are substantiated by their data, although I have suggestions for clarifying their analyses or providing caveats.

The work is novel and important as there are relatively few endophyte genomes available. I believe the manuscript would be of broad interest to both microbiologists, ecologists, and evolutionary biologists.

Below are my specific comments (in the order encountered).

Pg. 2, lines 47-48. The first sentence is missing references. Are you referring to mycorrhizal fungi here?

Pg. 2, line 50. The use of the word "retain" here implies evolutionary direction, but we don't know if endophytes retained this as an ancestral state or gained it. Change to "have the ability to colonize roots".

Pg. 2, lines 52-53. The sentence starting with "These fungal endophytes..." is a bit misleading as I would argue that stochastic interactions are not mutually exclusive of stable associations. Perhaps it is better to state that factors determining endophyte associations with hosts can be stochastic or deterministic? This language is also used on Pg. 3, line 68.

Pg. 2, Line 58. I suggest removing the phrase "evolution toward" since the how/when endophytism has evolved is complex. In some clades, endophytism may have been ancestral and may have transitioned to pathogenicity or saprotrophy (e.g., Arnold et al., 2009 Sys Bio).

Pg. 3, line 71. Define ITS for non-fungal crowd? Also, the wording in this sentence is awkward. I suggest revising to "3 phyla, 27 genera, and 37 species".

Pg. 3, line 81. Add a reference for root endosphere samples across European sites. Also, can you say how many plant species were included in this analysis?

Pg. 3, lines 87-89. I would remove the phrase "are not stochastic encounters" from this sentence as it is not needed.

Pg. 4, line 103. How do you define "close" phylogenetic relationships? Do you mean sister relationships? Also, it is difficult to judge this because the branch lengths are difficult to see on the radial tree. Can you provide the tree with branch lengths in rectangular form as a Supplement?

Pg. 4, lines 107-109. There is obviously a strong phylogenetic signal in the different gene categories (i.e., Fig. 2); thus I worry that lifestyle and phylogeny are confounded in the Random Forest analysis. Did you first quantify the degree of phylogenetic signal in the orthogroup presence/absence data? Please clarify.

Also, I realize that presence/absence of gene families is easiest to deal with, but I think it would be useful to also run the Random Forest with gene family abundance data. It may not be the complete presence/absence of orthogroups that explains phenotype, but rather the expansion of particular gene families.

Pg. 4, lines 111-112. Since genome databases are biased towards pathogens and lack endophytes (which is why this is such a valuable study!) it seems likely that your ASR results may falsely reconstruct ancestors as pathogens or saprotrophs. For example, *Colletotrichum* spp. are very common as foliar endophytes in the tropics and seed-associated fungi, although few genomes of these endophytes have been sequenced. I realize you are constrained by what genomes are available, but it is important the potential for this bias is discussed and to temper your conclusions about the evolution of these species due to this uncertainty.

Pg. 5, line 134. Can you justify here or in the methods why you choose to use Jaccard? Similar to your analysis of presence/absence of orthogroups, I think that using an abundance-based estimator is needed to the nuances of "endophytism".

Pg. 5, lines 135-137. Your sentence states that "lifestyle significantly explained variation..." but the stats in Extended Data Fig. 4 show that phylogeny always explains more variation than lifestyle. Thus, I think this sentence needs to be revised to reflect this information. Also, did you test to see whether there a significant interaction between lifestyle and phylo PC1/PC2 for permanova? Please clarify.

Also, in the next sentence you state "Interestingly, the factor "lifestyle" explained the highest percentage of variation for PCWDE repertoires..." but the reported stats contradict your statement (i.e., phylogeny $R^2 = 0.18$; lifestyle $R^2 = 0.16$). Please clarify.

Pg. 5, lines 142-143. It would be useful here to clarify what type of "beneficial endophytes" you are referring to here (i.e., do these all represent foliar endophytes?). It is unclear from Supplemental Table. Also, in what way are they beneficial? Citations to those studies would be

useful here.

It would be useful to state upfront that you were interested in knowing whether your root endophytes could be distinguished from other types of endophytes (i.e., here you provide details about what other endophytes are represented in your dataset). This would help guide the reader through this section.

These are very cool results! One question I have is whether your 41 root endophytes can also colonize leaves? Is this something you can check with GlobalFungi database? If similar ITS sequences have been isolated as foliar endophytes perhaps similar gene repertoires are important both for root and shoot colonization, which would be very interesting.

Pg. 6, line 165. It is unclear what published fungal transcriptomic datasets you are referring to here. Please clarify.

Pg. 6, line 169. I would revise sentence to "essential for endophytic root colonization" unless you provide more info on the "known endophytes".

Pg. 6, line 171. Since you mention the endophytic continuum, I would definitely cite Schulz and Boyle <https://pubmed.ncbi.nlm.nih.gov/16080390/>

Pg. 6, line 176. Here you state five week old plants, but legend for Figure says 4 week. Please clarify.

Pg. 7, lines 192-194. Previous studies have shown that endophytes of stressed plants are more likely to act as pathogens; thus, I'm surprised that you found more determinantal interactions with higher nutrients. Any ideas why?

As I know you are aware, measuring the impact of endophytes on plant performance using a single fungal isolate may not accurately reflect interactions of complex communities in nature. Thus, I think it would be very useful to re-examine the community data from Thiergart et al. 2020 to look at co-occurrence patterns between beneficial, neutral, and detrimental endophytes in roots. I know you found that detrimental endophytes had higher RA, but I wonder whether their impact on host could be neutralized by other endophytes.

Fig. 4. It is super interesting that most of the detrimental fungi are Sordariomycetes. Can you comment on this in Discussion?

Fig. 5A: It would be useful to show the class level information in the figure.

Fig. 5B. This Venn diagram is hard to read. Maybe an UpSet Plot would be better?

Pg. 7, line 211. Change "unrelated" to "distantly related".

Pg. 14, line 429. So, phylogenetic distances are just the sum of the branch lengths between each taxa? If so, I would say "pairwise phylogenetic distances (i.e., sum of the branch lengths)".

Lastly, I had a little trouble understanding all the intricacies of the statistical analyses based on what was written in the Results and Methods, but the github repo was very informative. I appreciate the authors making this information and code available so that their work is more easily understood and reproducible.

REVIEWER COMMENTS

Reviewer #1 (Remarks to the Author):

This is a very thorough work and a well written paper.

We thank the reviewer for the overall positive evaluation of our work.

I have only rather minor comments, and mostly regarding presentation and interpretation
1 the general assumption seems to be that (majority of) root-colonising fungi are culturable. To which degree this holds for endophytes? Clearly, for mycorrhizal fungi this is not true. The paper provides information on how the field-collected abundance and experimentally inoculated abundance of the focal fungal isolates (well and positively) correlate. But is there any caveats here, an aspect missed if non-culturable fungi are not considered?

We indeed focused here on the cultured fungi and we make it clear in the text (see lines 91, 93). Based on our recent work, we observed that around 50% of the most abundant ASVs (>1% RA) detected in *A. thaliana* roots have a representative isolate in our culture collections (please see Duran et al. 2018 *Cell*). We also showed here (see Fig. 1c) that the 41 fungi are representative of naturally occurring root mycobiomes but this set is of course not comprehensive enough. We have now included a sentence in the discussion (see lines 322-327) to insist on the fact that 1) many fungal taxa were not considered here, including endophytes that cannot be cultured and 2) that it remains to test whether the conclusions derived from our study with diverse 41 cultured isolates can apply (or not) for a broader range of fungal taxa.

2 the mycorrhizal fungi addressed in some analyses (genomics) include ectomycorrhiza, ericoid mycorrhiza and orchid mycorrhiza, but not arbuscular mycorrhiza. Why? Genomes of number of AMF are available. At minimum, it should be acknowledged when interpreting the data that AMF may be quite a different lot, genomically and genetically speaking

We would like to thank the reviewer for this comment. We decided not to include AMFs since they belong to a phylum that is very distant from our *Arabidopsis* mycobiota strains. We carefully selected the strains from the literature to reduce the phylogenetic signal in our dataset. The genomes we have included in our comparative genomics analysis were selected in the same fungal classes — or closely related ones — as *A. thaliana* mycobiota members. Including AMFs would have been possible but would have made it difficult to uncouple phylogeny from lifestyle effects in our comparative genomics. We have now included a sentence in the text (see lines 49-50) to explain why AMFs were not included.

L47: rich diversit - do you mean high diversity? Or rich set of?

We would like to apologize for the confusion. This community is both rich and diverse. We have now corrected this sentence (see line 48), and added references.

L176: the plant performance index combining both plant biomass and germination rate is very nice idea. How do the two components contribute to the general index, how variable are they? Any fungi influencing more the germination vs growth?

We would like to thank the reviewer for this comment. We have now added a new supplementary figure providing the germination rate of plants in response to seed inoculation with our 41 fungal strains (see Supplementary Figure 9). We observed that the negative effect of the fungi on plant performance is driven by both a reduction in the germination rate and a reduction of plant growth. This is why we took into account these two metrics to calculate a plant performance index. Interestingly, these new data suggest that some specific strains might have an important impact on seed germination (for instance, *Plectosphaerella cucumerina* 0016).

Fig 4: there is large variation in the effect of some fungi on plant performance. Is this a notable result? This is, fungi which effect is hard to predict, or potentially very context dependent (and hence, potentially could be managed/directed towards a desired outcome)?

Thanks for this remark. We are not sure whether the reviewer is referring to within-sample variation or between-sample variation. We have indeed identified few fungi (i.e. *Fusarium oxysporum* 0094, *Boeremia exigua* 0100) for which the effect on PPI is very variable between replicates. Note that these re-colonization experiments were performed three times independently (full factorial replicates) across multiple years and with multiple lab members involved. Since we have each time confirmed the identity of these fungi, we are confident that there was no contaminations or mix-up between strains. By replicating these fungi under artificial conditions, we cannot exclude the possibility that some fungi can rapidly change their developmental programs, maybe through epigenetic changes, which likely contributed to the experiment-to-experiment variation observed for some isolates.

Fig 4 a: what are the dashed lines indicating?

The dashed lines highlight the mean performance index of mock plants. This is now clearly stated in the figure legend (see lines 927-928).

Reviewer #2 (Remarks to the Author):

The manuscript by Mesny et al. provides a thorough assessment of the evolution of endophytism in fungi associated with roots of *Arabidopsis thaliana*, and the genomic signatures resulting from it. Focusing on a core set of root mycobiome members of that plant species, the authors re-analyzed existing amplicon sequencing data to evaluate their prevalence in roots across a wide geographic range, performed re-colonization experiments to test the effects of fungi on host growth, and did comparative genomics to identify genetic determinants of the endophytic lifestyle. The latter resulted in a list of candidate gene families best explaining the endophytic lifestyle, one of which was validated in recolonization experiments with existing mutants of a fungus over-expressing one target gene.

Overall, I think this is a comprehensive study that affords interesting insights into the functional ecology and plant interactions of a fungal group that, despite making up a major component of fungal communities in plants, have poorly understood associations with plants. Besides, it provides a very valuable dataset in the form of genome data from a fair number of fungi that are common root endophytes, not only of *Arabidopsis thaliana* but also from other plant hosts and habitats. Indeed, the main strength of the study lies in the amount of genomic data generated and the inferences made from them, whereas the data concerning the prevalence of fungi in roots and the effects on plant growth confirm results

from several previous studies. Several such studies are properly cited in the manuscript.

We thank the reviewer for the constructive feedback and overall positive evaluation of our work.

I only have a few comments about the manuscript, the first one dealing with its structure. Although the manuscript is well and clearly written, it does not provide upfront an specific explanation of the study background, aims, and hypotheses. Instead, the 'storyline' develops as the paper advances, so that specific questions are posited in the Results section just before being tackled. The Introduction is in fact very short and reads more like an extended abstract, with only a general exposé of the antecedents at the beginning (L47-56) followed by a summary of the research done (L56-64). Besides, specific aspects important for the research are not even introduced, as if the authors relied on the reader's knowledge of the relevant literature. For example, it is not explicitly explained why transposable elements (L125-126) were investigated, or why do some genes (L128-130) are expected to play important roles in fungal-host interactions rather than being common for, e.g., saprotrophy. Altogether, the way it is presented, the manuscript reads like an explorative study in which the authors did a lot of work and then looked for patterns in the results. I suggest the authors to clearly state upfront their motivation for doing this research, explicitly mentioning their specific aims and hypotheses or expectations, as is commonly requested for most scientific papers.

We would like to thank the reviewer for this comment. Indeed, the introduction was too short and the links with our research questions were not clearly detailed in the previous version of our manuscript. We have now extensively worked on the introduction section (see lines 48-88), included more background information regarding what is known in the literature (see lines 62-77), and more precisely defined our research strategy (see lines 77-88). We hope that the aspects raised by the reviewer have been addressed.

Many of the methodological approaches used are above my paygrade and thus I cannot judge their specifics, although in general they are explained well enough to understand what was done.

Thanks. We indeed tried to explain all the methods in detail and be as transparent as possible to allow other scientists to reproduce our data and to use the computational methods that have been employed here.

I do wonder about the ability to properly disentangle the effects of fungal lifestyles from those of phylogeny. Even though the authors selected quite a comprehensive list of fungal strains representing different life strategies (please, note that shoe-horning fungi into unique categories is often difficult; e.g. see Selosse et al. 2018. *New Phytologist* 217: 968–972.) and used different methods to account for phylogenetic signal (e.g. introducing phylogeny PCs in ANOVA tests or using PhyloGLM models), the phylogenetic representation of lifestyles is rather biased: e.g. with many closely related pathogenic members of the Nectriaceae, endophytes in the Helotiales, EcM in the Agaricales, and only a couple of related OcM (Fig. S2).

We tend to agree with the reviewer that grouping fungi according to their lifestyles is a potential weakness, not only of our work but also of the work of many others. We are aware of the debate and that's why we used multiple independent inputs to define the lifestyles (isolation source, description in original publication, the FunGuild database). We have modified the text to highlight this constraint and now cite the manuscript by Selosse

et al. 2018 (see lines 129-131). However, and as the reviewer mentioned just below, our strategy (careful strain selection, control for phylogenetic signal) is maybe the best we can do.

I acknowledge that the authors' efforts are probably the best that can be done, given the strong phylogenetic conservatism in some lifestyles. But my concerns lie on the capacity to statistically account for that conservatism. For example, do the letters above boxplots in Fig. 2b represent differences between lifestyles once phylogeny has been accounted for, or they represent instead differences by both phylogeny and lifestyle (this is not clear from the legend)?

We thank the reviewer for acknowledging our efforts at trying to analyze phylogeny-independent lifestyle effects on genome compositions. In the specific case of Figure 2b, we used PCA coordinates reflecting strains phylogeny in our statistical model. Since the residuals of the model were normally distributed, we could use ANOVA testing, and subsequently the post-hoc test Tukey Honestly Significant Difference. The advantage of using this specific post-hoc test is that it performs comparison between lifestyle groups, taking into account the variance that the "lifestyle" factor explains according to the ANOVA test. Therefore, if the 'lifestyle' factor significantly explains only a very small proportion of the dataset variance — according to the ANOVA test —, it is very unlikely that differences between categories will be identified. We have now added a sentence in the Methods section to better explain this TukeyHSD test (see lines 604-606).

In the case of Figure 2c (and Supplementary Table 3), we used a PERMANOVA test which is sequential: this test attributes the maximum possible variance to the first factor (phylogenetic PC1), then the maximum possible to the second one (PC2), etc. By putting 'lifestyle' as the last factor in the model formula, we ensure to "remove" the phylogenetic signal before to consider potential effects of lifestyle categories. Our observation that the composition in total orthogroups (see 'Total Proteins' in Figure 2c and Supplementary Table 3) is explained by phylogeny but not by lifestyle validates that approach. In contrast, the composition in PCWDE genes is showing the strongest lifestyle signal in our dataset suggests that our statistical approach can account for phylogenetic conservatism.

How would plots in Fig. S4 look if they were to show only the variance explained by lifestyle (e.g. 'vegan::dbRda(JaccardMatrix ~ Condition(Phylogeny) + Lifestyle)')? At any rate, in the relevant parts of the manuscript the authors should specify which amount of data variation is exclusively explained by lifestyle and phylogeny, and which amount cannot be differentiated between both factors (e.g. variation partitioning).

We would like to thank the reviewer for this suggestion. This supplementary figure (now Fig 6) is now presenting two dbRDA analyses per gene repertoire, respectively constrained by phylogeny and lifestyle. We used the suggested function to make these analyses (see lines 493-496 in the Methods section).

Additionally, we added a supplementary table (Supplementary Table 3) listing the R^2 and p -values obtained for the PERMANOVA analyses (see also lines 168-172 in the text).

My last comment refers to the relationship found between fungal abundance in roots and the detrimental effects on plant growth. This relationship makes sense and has indeed been shown in the past (see Kia et al. 2017. The ISME Journal 11: 777–790). However, here, the quantification of fungi in roots relied on qPCR with general fungal primers that target the rDNA ITS, a region with variable copy numbers across fungi (Lofgren et al. 2019. Molecular Ecology 28: 721–730) that would therefore distort direct comparisons

across species (other fungal traits, such as nuclei per cell, cells per hyphal length, etc. might also influence comparisons across species). Note also that, at least in bacteria, rDNA copy numbers are correlated with different life strategies (Klappenbach et al. 2000. *Applied and Environmental Microbiology* 66: 1328–1333.) or growth habits that could also affect, per se, interactions (again, see Kia et al. 2017). Some discussion about the validity and/or the caveats of the assessment should be included.

We would like to thank the reviewer for this comment. This is indeed an important point that we would like to clarify. Because of the broad phylogenetic diversity in our set of fungi, it was impossible to design a pair of primers that would lead to the amplification of a single target in each of our mycobiota genomes. We therefore decided to use the ITS1F/ITS2 primers, as they have been extensively used during the last two decades for estimating fungal relative abundance in natural communities.

As high-quality genome assemblies were obtained for our 41 fungi of interest, we inspected the link between qPCR amplification signal (Cq) and copy number of ITS sequences in genomes. To do so, we conducted a qPCR experiment with ITS1F and ITS2 primers on 24 mycobiota members with very different ITS copy numbers. We took care of using the same amount of fungal genomic DNA (1ng) as an input. We could not find any significant effect of ITS copy number on Cq values ($\text{lm}(\text{Cq} \sim \text{ITS copy number})$, ANOVA $P=0.9$).

We checked whether the (small) variation of Cq values identified by this experiment is revealing any primer bias that would introduce a skew in our colonization results. As shown by the plot below, there is no correlation between the mean plant colonization index we measured and the Cq values obtained on a same amount of the different fungal DNA extracts. We therefore believe that our colonization measurements by qPCR are biologically meaningful. However, to make the reader aware of this potential limitation, we included a sentence in the discussion to make it clear that we cannot exclude the possibility that copy number variation and primer bias might distort direct comparisons across species. We also cite Lofgren et al. 2019. *Molecular Ecology* 28: 721–730. (see lines 340-344).

Specific comments:

L75: rs -> please replace by 'rho', the standard representation of the Spearman's correlation coefficient (also in Fig. 1b and elsewhere in the text).

We thank the reviewer for identifying this error. "rs" has been replaced by "rho" in the manuscript and figures (see lines 103, 125-126, 220, 225-226 and Figures 1, 4, S2 and S10).

L76: n =30/41 -> does this refer to the number of samples, out of the total, in which RA was above 0.1%. Please, state clearly. Same for L77.

We would like to thank the reviewer to pinpoint that such notation is unclear. "n=30/41" indeed referred to the number of strains in which RA was above 0.1%. We have now replaced this notation by "30 out of 41" to make it less confusing (see lines 100-101).

L80-82: I have problems understanding the two panels in Fig. 1c. Please, describe more clearly.

To make this point more clear, we redesigned the Figure 1c to simplify it and make it self-explanatory. The figure now displays a clear comparison of the fungal biodiversity in *A. thaliana* roots covered by our previously-published ASV-approach (Thiergart et al. 2020) and by our selection of 41 isolates. We hope it is now easier to understand this panel. We have also re-written the figure legend (see lines 877-881).

L112: what are the grey sectors in the pie charts? Do they correspond to MyM? If that's the case, I wonder whether this category is comparable to the predefined lifestyles (SAP, EF, etc.): the latter summarize functional strategies of interaction with plant hosts, whereas MyM refers to the origin of the isolates (i.e. from roots of Arabidopsis). My point here is: do large fractions of gray in ancestral nodes indicate that ancestors were likely 'Arabidopsis mycobiota members'? The figure may be more clear if 'MyM' is replaced by 'unknown'.

We would like to thank the reviewer for this important comment, which helped us improving our analysis. We trained a new Random Forest classifier, this time only on the 79 non-mycobiota genomes, so the ancestral lifestyle predictions could only be lifestyles from the literature. As recommended by Reviewer #3, we used copy numbers of gene families instead of presence/absence. This new classification model has a better accuracy of $R^2 = 0.70$ (previously 0.56) (see new Fig 2a and lines 137-139). With this new approach, we only considered the most-reliable lifestyle categories of published genomes.

L112-115: the two examples here are not evident in Fig. 2a. Perhaps you could highlight in the figure the specific tree tips referred to in the main text.

We would like to thank the reviewer for this suggestion. We have added two numbered arrows pointing at the two ancestors which lifestyle was previously predicted in literature (see Fig. 2a and lines 145 and 888-891). It should now be easier to refer to the figure and identify these ancestors on the species tree.

L133: do letters above boxplots in Fig. 2b indicate differences exclusively based on lifestyle, or on both lifestyle and phylogeny? Please, indicate this explicitly, either here or in the figure legend (L828). I'd say that only differences purely based on lifestyle are relevant here.

These letters rely on a TukeyHSD test which was computed taking into account the variance explained by the Lifestyle factor in the previous ANOVA test. They are therefore purely relevant on lifestyle. We clarified this point in the Methods section. We used PCA coordinates reflecting strains phylogeny in our statistical model. Since the residuals of the model were normally distributed, we could use ANOVA testing, and subsequently the post-hoc test Tukey Honestly Significant Difference. The advantage of using this specific post-hoc test is that it performs comparison between lifestyle groups, taking into account the variance that the "lifestyle" factor explains according to the ANOVA test. Therefore, if the 'lifestyle' factor significantly explains only a very small proportion of the dataset variance — according to the ANOVA test —, it is very unlikely that differences between categories will be identified. We have now added a sentence in the Methods section to better explain this TukeyHSD test (see lines 604-606).

L297-300: please note that a similar relationship between root colonization aggressiveness and detrimental effects on host growth has been previously described (Kia. et al. ISME J 11, 777–790 (2017)).

Thanks. Great to see that another independent lab has previously observed this correlation. We have now added this reference at that lines (see lines 338-339).

L176: 'of five-week-old plants' -> '4-week-old plants' in legend of Fig. 4 (L887), and L484. Please, reconcile.

We thank the reviewer for pinpointing this mistake. We only grew plants for 4 weeks, and have corrected the text (see lines 211-212).

L430: wouldn't it be better to use PCoA than PCA to obtain the phylogenetic PCs? The former performs better in representing non-euclidean distances, such as phylogenetic distances (e.g. Desdevises et al. 2003. Evolution 57: 2647–2652.). In addition, why only

select the first two PCs? These might only collect the phylogenetic separation of a few major lineages, whereas additional PCs might provide a finer-grained representation of the phylogenetic relationships among fungi (a common approach involves variable selection for selecting relevant PCs). In any case, please provide the PCA/PCoA with the proportion of variance explained by the selected PCs.

We would like to thank the reviewer for raising this point. We have now incorporated two additional PCs in our testing (reaching a cumulative variance of 95%). We now present this PCA in a new Figure (see Supplementary Fig. 3). Other PCs represent less than 1% variance, and would have only introduced some noise in our analysis, since they very poorly represent the phylogenetic signal we are interested in. Interestingly, in our specific case, a PCoA represents less of the phylogenetic signal than a PCA (4 PCs only reach 72% of cumulative variance, see the graphs below). This might come from the types of between-organism distances calculated by the method 'STAG' implemented in OrthoFinder, from which we obtained our species tree.

L787: ...but based on...

L948: 'The first vertical stripe' -> The vertical stripe to the right?

L949: 'whereas the second' -> whereas the one to the left?

We would like to thank the reviewer for pointing these three errors, that we have now corrected (see lines 876, 956, 957).

L950: 'Gene family sizes and representation in the different lifestyles...' -> but is this an absolute measure? Wouldn't it be biased by the number of fungi included in each dataset? If that's the case, please normalize by genome (e.g. median size?)

We thank the reviewer for this comment. This measure is indeed absolute and biased towards the number of fungi in each lifestyle. However, with this barplot, we simply aimed at presenting

- 1) how big is each gene family across the whole genomic data set: we can here see that the pectate lyase is a big family, probably with a high number of copies in genomes
- 2) have a quick look at the representation of these gene families across different lifestyles.

The supplementary figure 14d better shows the different copy numbers of the Pectate lyase orthogroup, across the different lifestyles.

L1211, 1213: 'sp.' not in italics

We thank the reviewer for pointing this error. It has now been corrected (see Supplementary Information).

L1253, 1256: 'On the right' -> below?

We thank the reviewer for pointing this confusion, that we have now corrected (see Supplementary Information).

Reviewer #3 (Remarks to the Author):

Summary: Mesny et al., sequenced the genomes of 41 fungal endophytes previously isolated from Arabidopsis roots with the goal to identify a genomic basis for root endophytism. Arabidopsis endophytes were compared to 79 previously sequenced fungal genomes representing phylogenetically and ecologically diverse taxa. Comparative genomic analyses and machine learning methods identified a subset of genes that appear to be enriched in root symbionts and other endophytes. The authors also found that root endophyte impact on host plant performance varied from detrimental to beneficial, which they linked to the degree of root colonization. Fungi that were able to colonize more root tissue had expanded pectin gene repertoires and were more likely to be detrimental to host health.

Overall, I felt that the manuscript was well written, and the rationale, results, and implications of the work were clear. I believe the authors major claims are substantiated by their data, although I have suggestions for clarifying their analyses or providing caveats.

The work is novel and important as there are relatively few endophyte genomes available. I believe the manuscript would be of broad interest to both microbiologists, ecologists, and evolutionary biologists.

Many thanks for the great feedback and positive evaluation of our work.

Below are my specific comments (in the order encountered).

Pg. 2, lines 47-48. The first sentence is missing references. Are you referring to mycorrhizal fungi here?

Thank you for highlighting a lack of references associated to the first sentence of the manuscript. We have now added some to this revised version (see lines 48-49).

Pg. 2, line 50. The use of the word "retain" here implies evolutionary direction, but we don't know if endophytes retained this as an ancestral state or gained it. Change to "have the ability to colonize roots".

We would like to thank the reviewer for highlighting this important point. We have changed the phrasing as suggested (see line 52).

Pg. 2, lines 52-53. The sentence starting with "These fungal endophytes..." is a bit misleading as I would argue that stochastic interactions are not mutually exclusive of stable associations. Perhaps it is better to state that factors determining endophyte associations with hosts can be stochastic or deterministic? This language is also used on Pg. 3, line 68.

We would like to thank the reviewer for arguing on this terminology. Referring to the definition of stochasticity ("A stochastic process or system is connected with random probability", Cambridge English Dictionary), we believe that stochastic and stable associations are antonyms, since if a host-microbe interaction is stable, it is not randomly occurring and probably relies on strong genetic determinants.

Pg. 2, Line 58. I suggest removing the phrase "evolution toward" since the how/when endophytism has evolved is complex. In some clades, endophytism may have been ancestral and may have transitioned to pathogenicity or saprotrophy (e.g., Arnold et al., 2009 Sys Bio).

We thank the reviewer for pointing this other phrasing problem suggesting an evolutionary route to endophytism. We have rewritten our introduction and corrected an other occurrence of "evolution towards endophytism" in the Discussion (see line 315).

Pg. 3, line 71. Define ITS for non-fungal crowd? Also, the wording in this sentence is awkward. I suggest revising to "3 phyla, 27 genera, and 37 species".

We would like to thank the reviewer for these suggestions that we have implemented, and which make the method and sentences easier to understand (see lines 95-96).

Pg. 3, line 81. Add a reference for root endosphere samples across European sites. Also, can you say how many plant species were included in this analysis?

We have now added this reference (see line 105). Although the European transect study Thiergart *et al.* 2020 studied the microbiota of *A. thaliana* plants and surrounding grasses, we here only re-analysed data sampled from the single *A. thaliana* species.

Pg. 3, lines 87-89. I would remove the phrase "are not stochastic encounters" from this sentence as it is not needed.

We thank the reviewer for this comment. We have rephrased the sentence as suggested (see line 114-116).

Pg. 4, line 103. How do you define "close" phylogenetic relationships? Do you mean sister relationships? Also, it is difficult to judge this because the branch lengths are difficult to see on the radial tree. Can you provide the tree with branch lengths in rectangular form as a Supplement?

The published genomes we incorporated in our analysis were selected in the same phylogenetic classes as our *Arabidopsis* mycobiota members, or in neighboring ones. This is now clearly stated in the text (see lines 128-129). A rectangular version of the species tree has been added as a new supplementary figure (see Supplementary Fig. 3). We hope this makes the phylogeny of our dataset easier to read.

Pg. 4, lines 107-109. There is obviously a strong phylogenetic signal in the different gene categories (i.e., Fig. 2); thus I worry that lifestyle and phylogeny are confounded in the Random Forest analysis. Did you first quantify the degree of phylogenetic signal in the orthogroup presence/absence data? Please clarify.

We thank the reviewer for this comment. We think that removing or correcting for phylogenetic signal is important when it comes to identifying gene candidates, so such candidates are not biased or false. While we took care of correcting for phylogenetic signal when it came to identify genomic signatures for a lifestyle and candidate genes for pathogenicity, the present Random Forests classifier does not implement such correction on purpose. With this Random Forests classifier, we only aim at obtaining the best predictive model for lifestyles (highest R^2), which can only be reached by including phylogenetic signal, since phylogeny is determinant of one fungus' lifestyle. Although it is impossible to assemble a perfectly balanced data set, we would like to highlight that such approach was made possible because we took care of including genomes from fungi with diverse lifestyles in every phylogenetic class.

Also, I realize that presence/absence of gene families is easiest to deal with, but I think it would be useful to also run the Random Forest with gene family abundance data. It may not be the complete presence/absence of orthogroups that explains phenotype, but rather the expansion of particular gene families.

We would like to thank the reviewer for this important comment which helped us improving our analysis. We have now trained a new Random Forest classifier, only on the 79 non-mycobiota genomes, as suggested by Reviewer #2. This time, we used copy numbers of gene families instead of presence/absence, using a Wagner parsimony approach instead of a Sankoff parsimony one (see lines 453-457 in the Methods section). This new classification model has a better accuracy of $R^2 = 0.70$ (previously 0.56). (see lines 137-139).

Pg. 4, lines 111-112. Since genome databases are biased towards pathogens and lack endophytes (which is why this is such a valuable study!) it seems likely that your ASR results may falsely reconstruct ancestors as pathogens or saprotrophs. For example, *Colletotrichum* spp. are very common as foliar endophytes in the tropics and seed-associated fungi, although few genomes of these endophytes have been sequenced. I realize you are constrained by what genomes are available, but it is important the potential for this bias is discussed and to temper your conclusions about the evolution of these species due to this uncertainty.

We would like to thank the reviewer to raise this important point on which we do agree. Genome databases are unfortunately biased, as well as all possible comparative genomics data set. As previously mentioned, we took care of including genomes from fungi with diverse lifestyles in every phylogenetic class covered by our study. However, it is still biased towards the genomes that are available, the ones we picked, and what is published about them (for lifestyle classification). Therefore, we added one sentence tempering our lifestyle predictions, referring to this data set bias (see lines 149-150), and

also sentences about how we picked our genomes (see lines 128-129) and classified them into lifestyles (see lines 130-131).

Pg. 5, line 134. Can you justify here or in the methods why you choose to use Jaccard? Similar to your analysis of presence/absence of orthogroups, I think that using an abundance-based estimator is needed to the nuances of "endophytism".

We thank the reviewer for this comment. Jaccard dissimilarity indices between genomes were calculated on family copy numbers in gene repertoires, not on family presence/absence. Since these count variables are discrete, Jaccard is an adapted type of distance to use in this context. We have now added an explanation for the choice of Jaccard in the Methods section (see lines 486-487).

Pg. 5, lines 135-137. Your sentence states that "lifestyle significantly explained variation..." but the stats in Extended Data Fig. 4 show that phylogeny always explains more variation than lifestyle. Thus, I think this sentence needs to be revised to reflect this information. Also, did you test to see whether there a significant interaction between lifestyle and phylo PC1/PC2 for permanova? Please clarify.

We thank the reviewer for this comment. Although phylogeny explains more variation than lifestyle, the PERMANOVA analysis identified the factor 'lifestyle' to significantly contribute in the variation of composition ($p < 0.05$). We have rephrased this sentence to remove potential ambiguity on this point (see lines 169-170).

The PERMANOVA analysis was re-run with additional PCs (as suggested by reviewer #2) and taking phylogeny:lifestyle interactions into account. The detailed results of these analyses can now be found in a new supplementary table (see Supplementary Table 3).

Also, in the next sentence you state "Interestingly, the factor "lifestyle" explained the highest percentage of variation for PCWDE repertoires..." but the reported stats contradict your statement (i.e., phylogeny $R^2 = 0.18$; lifestyle $R^2 = 0.16$). Please clarify.

Here, the word 'highest' does not refer to the comparison between the phylogeny and lifestyle factors, but to the comparison of variance explained by lifestyle between the different gene repertoires. It remains correct to say that PCWDE-encoding genes correspond to the gene category for which the signal "Lifestyle" was the strongest (i.e. compared to the other gene categories tested). We made it clearer in the text (see lines 170-172).

Pg. 5, lines 142-143. It would be useful here to clarify what type of "beneficial endophytes" you are referring to here (i.e., do these all represent foliar endophytes?). It is unclear from Supplemental Table. Also, in what way are they beneficial? Citations to those studies would be useful here.

It would be useful to state upfront that you were interested in knowing whether your root endophytes could be distinguished from other types of endophytes (i.e., here you provide details about what other endophytes are represented in your dataset). This would help guide the reader through this section.

We would like to apologize for the confusing phrasing of this sentence. Here we looked at total endophytes from our comparative genomic dataset, not at a subset of them. We have now removed the "beneficial" adjective from this sentence (see line 174). Because of an important lack of fungal endophytes in genomic databases, our genomic data set comprises 9 fungal endophytes: 8 originating from roots and 1 from leaves (see

Supplementary Table 2). The references associated to each of these endophytic strains can be found in Supplementary Table 2 (PMIDs) and are also cited at line 132. The potential beneficial effect these endophytes might have on their host is not considered here. We are not interested in comparing our mycobiota members to different types of endophytes, but to see if their composition in genes known to be involved in fungal association to host or environment resemble the ones of another lifestyle. It turns out it resembles the ones of fungi previously-described as endophytes.

These are very cool results! One question I have is whether your 41 root endophytes can also colonize leaves? Is this something you can check with GlobalFungi database? If similar ITS sequences have been isolated as foliar endophytes perhaps similar gene repertoires are important both for root and shoot colonization, which would be very interesting.

We would like to thank the reviewer for this interesting suggestion. We have now added a new supplementary figure which shows the prevalence and abundance of the taxa we studied in leaves, based on Agler *et al.* 2016 and GlobalFungi data. This new analysis revealed that only a small portion of these 41 root associated fungi (<30%) is detected in *A. thaliana* leaves at two locations in Germany and in leaf samples from diverse plant species worldwide (see lines 111-113 and new Supplementary Fig. 1).

Pg. 6, line 165. It is unclear what published fungal transcriptomic datasets you are referring to here. Please clarify.

We thank the reviewer for this comment. We improved the phrasing of this sentence to be more safe-explanatory (see line 198). Here was used the total set of fungal transcriptomic datasets that are comprised in the database STRING to emit co-expression indices.

Pg. 6, line 169. I would revise sentence to "essential for endophytic root colonization" unless you provide more info on the "known endophytes".

We would like to thank the reviewer for this suggestion, that we now took into account (see lines 202).

Pg. 6, line 171. Since you mention the endophytic continuum, I would definitely cite Schulz and Boyle <https://pubmed.ncbi.nlm.nih.gov/16080390/>

We would like to thank the reviewer for this suggestion. We now cite this publication (see line 205).

Pg. 6, line 176. Here you state five week old plants, but legend for Figure says 4 week. Please clarify.

We thank the reviewer for pointing this mistake. We only grew plants for 4 weeks. This has been corrected in the text (see lines 210-212).

Pg. 7, lines 192-194. Previous studies have shown that endophytes of stressed plants are more likely to act as pathogens; thus, I'm surprised that you found more determinantal interactions with higher nutrients. Any ideas why?

This is an interesting hypothesis. Our previous work on *C. tofieldiae* revealed that this fungal endophyte actually better promoted plant growth when P levels are low, which is

consistent with what we observed here (Hiruma et al. 2016 Cell, Hacquard et al. 2016 Nature Communications). We have not inspected the link here between fungal effect on PPI under LowPi/HighPi and innate immunity. Although it is possible that under LowPi, plants invest into the P starvation response at the expense of defense (as previously shown for bacteria in Castrillo et al. 2018), the interplay between fungal endophytes, innate immunity and PSR is less clear. In nature, plants are often facing LowPi conditions so the mechanism(s) linking P levels in soil and increased fungal pathogenicity (if it exists) might not be that simple. An alternative explanation is that most of these endophytes (including beneficial and detrimental) can promote (at least to some extent) host P uptake when P levels are extremely low.

As I know you are aware, measuring the impact of endophytes on plant performance using a single fungal isolate may not accurately reflect interactions of complex communities in nature. Thus, I think it would be very useful to re-examine the community data from Thiergart et al. 2020 to look at co-occurrence patterns between beneficial, neutral, and detrimental endophytes in roots. I know you found that detrimental endophytes had higher RA, but I wonder whether their impact on host could be neutralized by other endophytes.

We would like to thank the reviewer for this interesting comment. As a new supplementary figure, we now provide a co-occurrence matrix of the 41 fungal taxa, calculated with the abundance data from Thiergart *et al.* 2020. This heatmap (see new Supplementary Fig. 11) reveals a conserved set of core-taxa which includes some fungi we identified to be detrimental. Consistently with our results suggesting a correlation between colonization ability and detrimental potential (see Fig 4), the strains we identified as beneficial for plant growth do not colonize plant extensively and are therefore rarely detected in natural microbial community. It is however possible that neutral strains in this core-set of co-occurring strains neutralize the effects of detrimental strains. However, we have evidence from a previous study (Duran *et al.*, 2018 - Cell) that *A. thaliana* bacterial root commensals are necessary for plant survival in presence of the mycobiota, which is pathogenic when inoculated as a synthetic community. Therefore, we believe that both bacteria and fungi are likely to contribute in restraining the detrimental effects we identified in mono-association here — and in a synthetic community in Duran et al., 2018.

Fig. 4. It is super interesting that most of the detrimental fungi are Sordariomycetes. Can you comment on this in Discussion?

Dothideomycetes and Sordariomycetes classes contain multiple strains with detrimental activities. This is consistent with the idea that they evolved from pathogenic ancestors (see lines 145-148, and Fig. 2a). Using the same lifestyle predictor we used for Figure 2a, we could predict the lifestyle of our 41 mycobiota members (see new pie charts on Figure 4), and saw that both these phylogenetic classes have a high pathogenic potential, in contrast with Leotiomycetes and Agaricomycetes.

In the last part of our study (see lines 206-306, and Fig. 6, we could link the high number of detrimental fungi at low Pi in Sordariomycetes to the copy number of gene family pectate lyase PL1_7. As shown by figure 5c, this family is over-represented in Sordariomycetes. We do believe that this family contributes in the detrimental effects observed in this phylogenetic class.

Fig. 5A: It would be useful to show the class level information in the figure.

We would like to thank the reviewer for this suggestion. We have added the phylogenetic classes in the legend of Figure 5a, and hope this makes the figure easier to read.

Fig. 5B. This Venn diagram is hard to read. Maybe an UpSet Plot would be better?

We would like to thank the reviewer for this suggestion. UpSet plots are indeed better for analysis purposes. We created one that shows the information of Figure 5b differently:

However, because of the high number of possible combinations with the 6 fungi we study, we decided to keep presenting these data as a Venn diagram because the results we present focus on the gene families that are commonly over-expressed in the six fungi. This number is better shown and easier to read on our original Venn diagram.

Pg. 7, line 211. Change "unrelated" to "distantly related".

We thank the reviewer for this comment. This has now been corrected (see line 249).

Pg. 14, line 429. So, phylogenetic distances are just the sum of the branch lengths between each taxa? If so, I would say "pairwise phylogenetic distances (i.e., sum of the branch lengths)".

We thank the reviewer for this comment. We have implemented this suggestion (see line 476).

Lastly, I had a little trouble understanding all the intricacies of the statistical analyses based on what was written in the Results and Methods, but the github repo was very informative. I appreciate the authors making this information and code available so that their work is more easily understood and reproducible.

We thank the reviewer for this comment. Both sections have been improved and an additional supplementary table has been added (PERMANOVA results on Supplementary Table 3) to make the results more understandable.

REVIEWERS' COMMENTS

Reviewer #1 (Remarks to the Author):

I'd like to thank the authors for careful revision. I find that all my earlier comments have been properly addressed. I have no further comment

Reviewer #2 (Remarks to the Author):

I thank the authors for the consideration they've given to my comments and their clear responses to all of my critiques, as well as to those by the other reviewers. Overall, I am satisfied about how my concerns have been addressed, and so I have no further comments. Again, I commend the authors for their thorough study.

Jose G. Maciá-Vicente

Reviewer #3 (Remarks to the Author):

[No further comments for authors]

Reviewer #1 (Remarks to the Author):

I'd like to thank the authors for careful revision. I find that all my earlier comments have been properly addressed. I have no further comment

THANKS

Reviewer #2 (Remarks to the Author):

I thank the authors for the consideration they've given to my comments and their clear responses to all of my critiques, as well as to those by the other reviewers. Overall, I am satisfied about how my concerns have been addressed, and so I have no further comments. Again, I commend the authors for their thorough study.

Jose G. Maciá-Vicente

THANKS

Reviewer #3 (Remarks to the Author):

[No further comments for authors]

Not applicable